# Checkpoint inhibition through small molecule-induced internalization of programmed death-ligand 1

Jang-June Park [1], Emily P. Thi[1], Victor H. Carpio[1], Yingzhi Bi[1], Andrew G. Cole[1], Bruce D. Dorsey[1], Kristi Fan[1], Troy Harasym[1], Christina L. Iott[1], Salam Kadhim[1], Jin Hyang Kim [1], Amy C. H. Lee[1], Duyan Nguyen[1], Bhavna S. Paratala[1], Ruiqing Qiu[1], Andre White [2], Damodharan Lakshminarasimhan[2], Christopher Leo[2], Robert K. Suto[2], Rene Rijnbrand[1], Sunny Tang[1], Michael J. Sofia[1] & Chris B. Moore [1✉]

Programmed death-ligand 1 is a glycoprotein expressed on antigen presenting cells, hepatocytes, and tumors which upon interaction with programmed death-1, results in inhibition of antigen-specific T cell responses. Here, we report a mechanism of inhibiting programmed death-ligand 1 through small molecule-induced dimerization and internalization. This represents a mechanism of checkpoint inhibition, which differentiates from anti-programmed death-ligand 1 antibodies which function through molecular disruption of the programmed death 1 interaction. Testing of programmed death ligand 1 small molecule inhibition in a humanized mouse model of colorectal cancer results in a significant reduction in tumor size and promotes T cell proliferation. In addition, antigen-specific T and B cell responses from patients with chronic hepatitis B infection are significantly elevated upon programmed death ligand 1 small molecule inhibitor treatment. Taken together, these data identify a mechanism of small molecule-induced programmed death ligand 1 internalization with potential therapeutic implications in oncology and chronic viral infections.

[1] Arbutus Biopharma Inc, Warminster, PA, USA. [2] Xtal BioStructures Inc., Natick, MA, USA. ✉email: cmooreERC@gmail.com

Programmed cell death-ligand 1 (PD-L1), also known as B7 homolog 1 or CD274 belongs to the B7 family of immunoregulatory ligands, and is widely expressed by lymphoid and nonlymphoid tissues[1–3], as well as various tumors[4–7]. PD-L1 expressed by tumor cells plays a critical role in the induction of inhibitory signals through the interaction with programmed cell death-1 (PD-1) expressed on the cell surface of T cells. This PD-1/PD-L1 interaction results in the suppression of tumor-specific T cell responses functioning as a tumor immune evasion mechanism[2,3,8]. Immune checkpoint inhibitors targeting the PD-1/PD-L1 interaction have become a successful immunotherapy in treating many advanced cancers[9–11] and are based on a mechanism of monoclonal antibody binding to and directly disrupting the PD-1/PD-L1 interaction[9,12,13]. Three such αPD-L1 antibodies have been approved by the FDA: atezolizumab, durvalumab, and avelumab for the treatment of bladder cancer, non-small cell lung carcinoma, and Merkel cell cancer and another candidate antibody, BMS-936559 is currently under clinical investigation[10,14]. These antibodies were reported to possess high affinities ranging from 40 to 700 pM and co-crystallography studies have revealed a mechanism of direct binding to the PD-1/PD-L1 interface[15–19]. To date, the discovery of small-molecule PD-1/PD-L1 inhibitors has lagged primarily due to the large interacting surface area and tight binding of the PD-1/PD-L1 interaction. It has been postulated that the discovery of a PD-1/PD-L1 small-molecule inhibitor would require these molecules to inhibit through a completely different mechanism of action. Nonetheless, the discovery of a PD-L1 small-molecule checkpoint inhibitor could dramatically advance the field of checkpoint modulation and possibly lead to the development of therapeutics with better clinical properties than current antibody treatments[20–22]. Recent reports describing the crystal structure of small-molecule/PD-L1 complexes have suggested a possible binding mode involving a hydrophobic pocket between two antiparallel PD-L1 molecules inducing the dimerization of two soluble PD-L1 molecules[8,23]. While this in vitro cell-free dimerization of two PD-L1 molecules could function to putatively mask the PD-1 interaction site, to date, this has not been described as a bonafide biological mechanism in vivo[23].

Syngeneic tumor mouse models are well established and frequently used to demonstrate in vivo efficacy profiles for checkpoint inhibitors. Further development of mouse knock-in models capable of expressing human PD-1 or PD-L1 has in turn enabled evaluation of inhibitors with specific activity against human checkpoint targets[24–28]. Previous studies have demonstrated that these models recapitulate elements of the tolerogenic tumor microenvironment present in patients, which are mediated to a large degree by immune checkpoint mechanisms, such as the PD-1/PD-L1 axis[24,29,30]. Antibody blockade of PD-1 or PD-L1 has been shown to be beneficial in reducing tumor volumes and mediating antitumor immune responses in these humanized PD-1/PD-L1 mouse models, which parallels effects observed in patients[25,26]. In addition, there is accumulating evidence that viral infections can also upregulate immune checkpoints, such as PD-1/PD-L1 as a mechanism to evade host immunity[31–33]. In fact, it has been proposed that immunotherapy targeting checkpoints could be used to break viral antigenic immune tolerance, resulting in the potential for restoration of natural immunity and clearance of infected cells[32,34–37]. One viral infection with the clear development of antigenic tolerance is chronic hepatitis B (CHB)[38]. In CHB patients, hepatitis B virus-specific T and B cells are dampened, in part due to high antigenemia and upregulation of PD-1 (T and B cells) and PD-L1 (hepatocytes and APCs). Accordingly, treatment of CHB patient peripheral blood mononuclear cell (PBMC) with antibody checkpoint inhibitors results in partial restoration of HBV-specific T and B cell responses[31,39,40]. Here, we report a mechanism for small-molecule inhibition of the PD-1/PD-L1 axis with potential clinical benefit in oncology and chronic viral infections.

## Results

**Identification of a potent small-molecule PD-L1 inhibitor.** Small molecules were first screened in a cell-free in vitro homogeneous time-resolved fluorescence (HTRF) assay, utilizing αPD-L1 (MIH1) and αPD-1 (nivolumab) antibodies as positive controls. A compound identified as ARB-272572 (compound A) inhibited PD-1/PD-L1 HTRF at 400 pM IC50 compared to 200 pM (αPD-L1) and 200 pM (αPD-1). Another compound of similar chemotype identified as ARB-276309 (compound B) was inactive in the HTRF assay (Fig. 1A). It has been reported that a small molecule can stabilize a dimer form of PD-L1 through interactions with antiparallel residues in the hydrophobic dimer interface[23]. To test if compound A can induce intermolecular PD-L1 interactions in solution, we developed a homodimer HTRF assay. Compound A, but not αPD-L1 (atezolizumab) or αPD-1 (nivolumab) induced PD-L1 protein intermolecular interactions thereby increasing the HTRF fluorescent signal, indicating compound A can induce PD-L1 homodimeric interactions (Fig. 1B). In addition, we were able to visualize this solution-based compound A-induced PD-L1 dimerization through crystallographic studies (Supplemental Fig. 2B). In order to extend these findings to a cell-based system, compound A was tested in a Jurkat/CHO-PD-1/PD-L1 bioassay. In this assay, preventing the PD-1/PD-L1 interaction results in a release of constitutive NFAT reporter suppression and increases luciferase activity (relative luminescence units). As expected, αPD-L1 and αPD-1 antibodies resulted in the potent cellular inhibition of PD-1/PD-L1 at 300 and 600 pM IC50s, respectively (Fig. 1C). Compound A was also active in this cellular assay (17 nM IC50), suggesting this small molecule was capable of inhibition of PD-1/PD-L1 in cells (Fig. 1C). We determined if compound A cellular bioassay potency could translate into immunomodulatory effects on primary human immune cells. It is known that αPD-1 antibodies can upregulate cytomegalovirus (CMV)-specific T cell responses[41]. Accordingly, we tested compound A in a CMV recall assay (Fig. 1D). In this assay, PBMCs are collected from CMV-seropositive subjects and stimulated with CMV antigens followed by measurements of IFNγ release by ELISA. Interestingly, both compound A and αPD-L1 antibodies exhibit similar upregulation of CMV responses, 3 and 0.7 nM IC50s, respectively[9] (Fig. 1D). In addition, nonspecific activation of the Jurkat PD-1 luciferase or primary immune cells by compound A was not observed (Supplemental Figs. 3C and 4A). Taken together, these data suggest compound A is a potent in vitro inhibitor of PD-1/PD-L1 cell signaling through induction of PD-L1 protein homo-interactions, which can lead to immunostimulatory activity in human primary cells.

**Small-molecule-induced PD-L1 dimerization and internalization.** We observed compound A-induced PD-L1 homodimeric interactions in solution (Fig. 1B). Next, we sought to determine if compound A could modulate native PD-L1 protein in living cells. To test this, we treated PD-L1 expressing CHO cells with compound A for 1 h followed by detection of PD-L1 by native gel electrophoresis. Within 1 h, compound A induced a shift to a larger molecular weight, doubling the observed monomeric form (Supplemental Fig. 3B). To test if compound A is simply inducing a post-lysis dimerization in solution, we treated detergent free lysates with compound A and visualized PD-L1 protein by native gel electrophoresis. We saw no post-lysis dimerization suggesting that in living cells an intact plasma membrane is required for

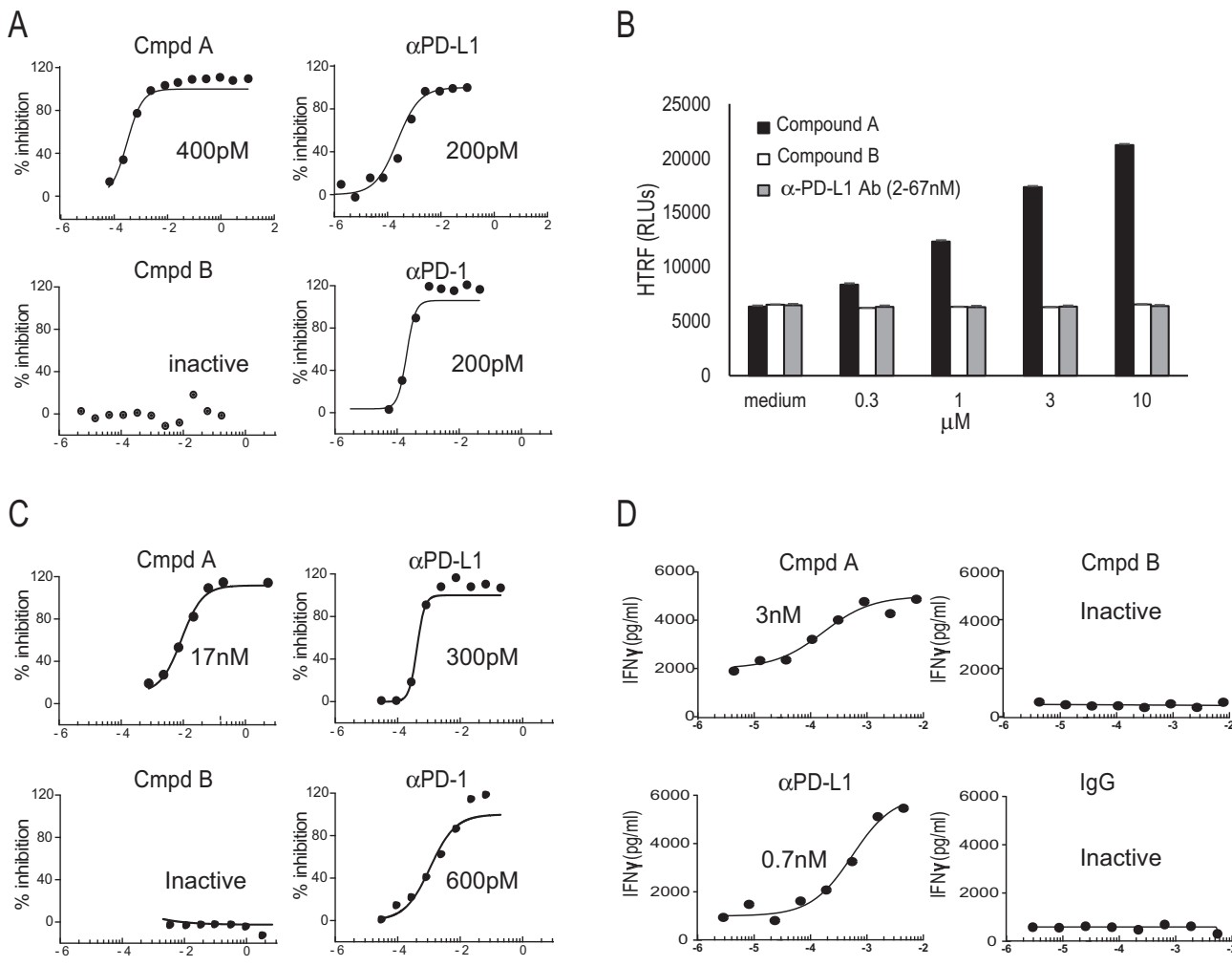

**Fig. 1 In vitro and ex vivo inhibition of PD-1/PD-L1 interaction. a** PD-1/PD-L1 homogenous time-resolved fluorescence (HTRF) assay. **b** PD-L1/PD-L1 HTRF dimerization assay. **c** PD-1/PD-L1 NFAT reporter bioassay. **d** CMV recall assay: IFNγ production measured by ELISA following stimulation of PBMCs with CMV antigens. Source data are provided as a Source data file.

compound A-induced PD-L1 dimerization. We can only surmise that this is because unlike the buffer-based cell-free dimerization observed in Fig. 1B, PD-L1 proteins need to be in close proximity on the plasma membrane to induce homodimer formation, and detergent free lysates are simply too viscous to allow for post-lysis PD-L1 dimerization. Nonetheless, we believe these data indicate that compound A dimerization is indeed occurring on the cell membrane in living cells.

Ligand-induced dimerization has been reported for other cell surface receptors. In addition, receptor dimerization often triggers downstream signaling events or receptor internalization[42,43]. Interestingly, 1 h treatment of compound A results in an almost complete loss of cell surface PD-L1 as measured by flow cytometry (Fig. 2C), in contrast to the effect of αPD-1/αPD-L1 antibodies or treatment with an inactive small molecule (compound B), in which PD-L1 was still observed on the cell surface. When PD-L1 expression was visualized by confocal microscopy, we observed a typical ring-like staining pattern indicative of exclusive cell membrane localization (Fig. 2D). However, upon 1 h treatment with compound A, PD-L1 rapidly internalized into the cytosol and was observed within cytosolic punctate structures (Fig. 2D and Supplemental Fig. 4C). PD-L1 protein–protein interaction occurred within 10 min of compound A treatment (Fig. 2E) and full cytosolic uptake of cell surface PD-L1 was complete within 1 h

(Fig. 2D). These data suggest that compound A-induced dimerization likely occurs prior to internalization. In addition, we were able to significantly inhibit compound A-induced PD-L1 internalization following pretreatment exposure of CHO-PD-L1 expressing cells to 4 °C (Supplemental Fig. 3A). Both internalization ($R^2 = 0.690$) and loss of PD-L1 cell surface expression ($R^2 = 0.580$) significantly correlate with the bioassay potency of small-molecule inhibitors of the same compound A chemotype, suggesting PD-L1 internalization is the primary mechanism contributing to the observed compound A cellular potency (Supplementary Fig. 1A, B).

Finally, we explored the reversibility of compound A-induced internalization. To test this, we treated PD-L1 aAPC/CHO-PD-L1 cells with compound A for 1 h, then washed away the compound and measured levels by liquid chromatography–tandem mass spectrometry (LC-MS/MS). After compound A was removed, we observed a precipitous drop in compound A concentration to the limit of detection by 6 days post treatment (Fig. 2F). Once compound reached undetectable levels, there was a full reconstitution of cell surface PD-L1 that reached pretreatment levels within 48 h (Fig. 2F). Treatment with 5 µg/ml actinomycin D or a golgi plug protein transport inhibitor prevented PD-L1 reconstitution, suggesting that transcription and new protein transport are required for replenishment of cell surface PD-L1 (Fig. 2G).

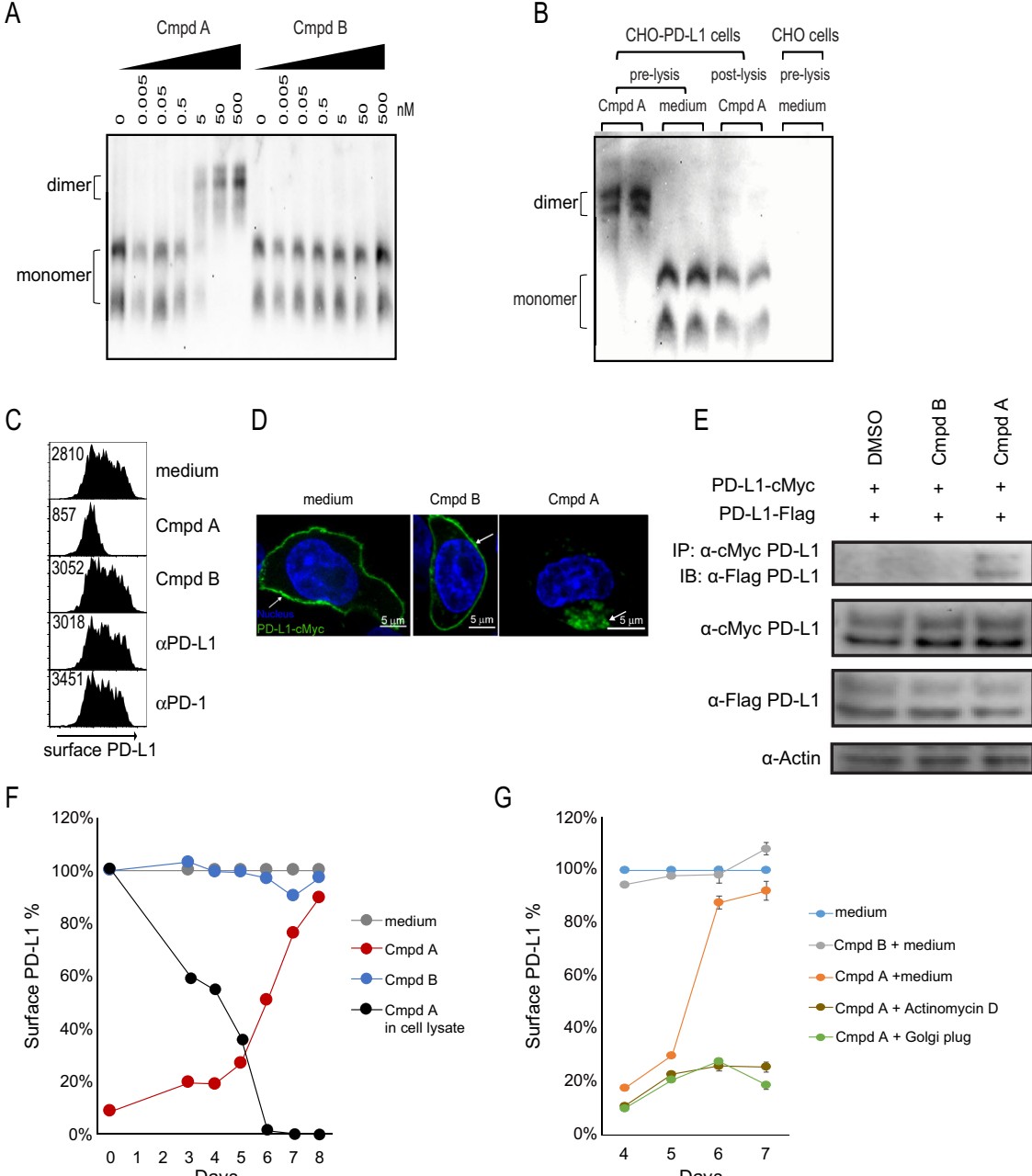

**Fig. 2 Mechanism of action of compound A. a** PD-L1 aAPC/CHO-K1 cells were treated with 0–500 nM compound A or compound B. PD-L1 protein visualized by native gel electrophoresis. **b** Native gel electrophoresis of PD-L1 protein following pre- and post-lysate treatment with compound A. **c** PD-L1 aAPC/CHO-K1 cell line was treated with compound A, αPD-1/αPD-L1 antibodies, or no treatment followed by cell surface PD-L1 detected by flow cytometry. Values inside histogram represent GeoMean fluorescence intensity. **d** CHO-K1 cells transfected with cMyc-PD-L1 and labeled with anti-cMyc Alexa Fluor 488-conjugated antibody followed by incubation with compound A. The confocal fluorescence microscopy detects PD-L1-cMyc (green) and nucleus (blue). Images are representative of three experiments and a minimum of 50 cells observed in each experiment. **e** Whole-cell extracts of CHO-K1 cells co-transfected with cMyc-PD-L1 and Flag-PD-L1 were incubated with media + 0.5% DMSO, inactive (cmpd B) or active (cmpd A) compound, and subjected to immunoprecipitation (IP) with anti-cMyc magnetic beads and immunoblot with anti-Flag antibody. Blots are representative of three experiments. **f** PD-L1 aAPC/CHO-K1 cell line treated with compound A to allow for loss of cell surface PD-L1, followed by daily media washes. Black line represents diminishing concentration of compound A as measured by LC-MS. Red line represents reconstitution of cell surface PD-L1 as measured by flow cytometry of ≥10,000 cells per time point. **g** Compound A wash off and PD-L1 cell surface reconstitution in the presence of transcriptional inhibitor (actinomycin D) and protein transport inhibitor (golgi plug). Inactive compound B (gray line) referenced as negative control. Source data are provided as a Source data file.

**Antitumor effects of small-molecule PD-L1 inhibition.** Antitumor efficacy of compound A treatment was evaluated in mice expressing humanized PD-1 and PD-L1, in which the extracellular domains of murine PD-1 and PD-L1 were genetically replaced with the human extracellular domain counterparts[25–27].

In these humanized PD-1/PD-L1 mice implanted with the colorectal cancer cell line MC38, compound A treatment at 10 mg/kg for only 7 days resulted in 60.4% group mean tumor volume reduction (relative to vehicle control) with four out of eight animals exhibiting response similar to those observed in the

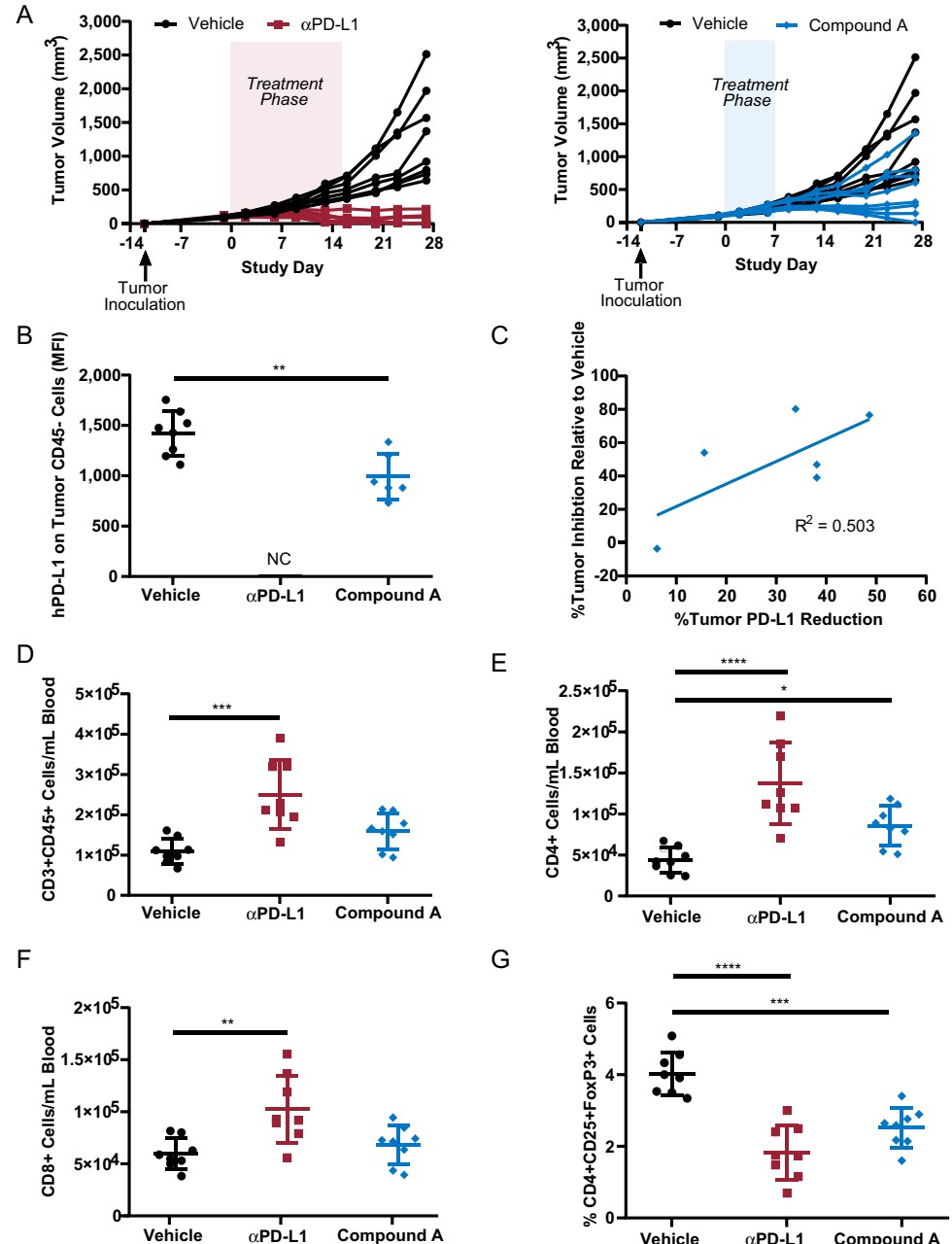

**Fig. 3 PD-L1 targeting in mouse model for colorectal cancer. a** Tumor growth of PD-1/PD-L1 humanized mice implanted with MC38 cells expressing humanized PD-L1 and treated with compound A (10 mg/kg orally for seven daily doses days 0–6), or αPD-L1 (10 mg/kg intraperitoneal injection thrice weekly days 0–16). Statistically significant tumor growth inhibition was observed with αPD-L1 antibody or compound A treatment ($p < 0.0001$ or $p = 0.009$, respectively, group means comparison relative to vehicle by one-way ANOVA with Dunnett's multiple comparisons test). **b** Compound A treatment mediates decrease in tumor PD-L1 expression at day 28. $**p = 0.002$ by unpaired $t$ test. NC data not collected due to insufficient tumor amounts for analysis. **c** Tumor inhibition is directly correlated with the degree of tumor PD-L1 reduction. **d–f** Counts of CD3[+] cells in blood at day 28, with further gating for CD4[+] or CD8[+] cells. **g** Frequency of T regulatory cells in blood. $*p = 0.03$, $**p = 0.003$, $***p = 0.0002$, $****p < 0.0001$ by one-Way ANOVA with Dunnett's multiple comparisons test. Source data are provided as a Source data file.

atezolizumab reference group, including one animal with undetectable tumor at study termination (study day 28; Fig. 3A). These inhibitory effects on tumor growth were accompanied by a decrease in the expression of PD-L1 in CD45[−] cells in the tumors, consistent with the anticipated mechanism of action of this compound, which promotes loss of cell surface PD-L1 (Fig. 3B). The decrease in PD-L1 expression in tumor cells directly correlated with the degree of tumor inhibition, with animals with the highest levels of PD-L1 reduction in tumors associated with the greatest tumor inhibitory response (Fig. 3C). PD-L1 reduction also

coincided with the presence of compound A in tumors (Supplemental Fig. 5). At study termination, compound A treatment resulted in a $1.4 \pm 0.4$-fold increase in circulatory CD3[+] cells, which further comprised of significant $2.0 \pm 0.6$-fold increase in CD4[+] T cells. No significant difference in total CD8[+] T cells was observed at study termination (study day 28 with compound A treatment (Fig. 3D–F)). In addition, a significant reduction in circulating T regulatory cells was also observed with compound A treatment (Fig. 3G). These effects on circulating T cells by compound A treatment were similar to changes observed in animals

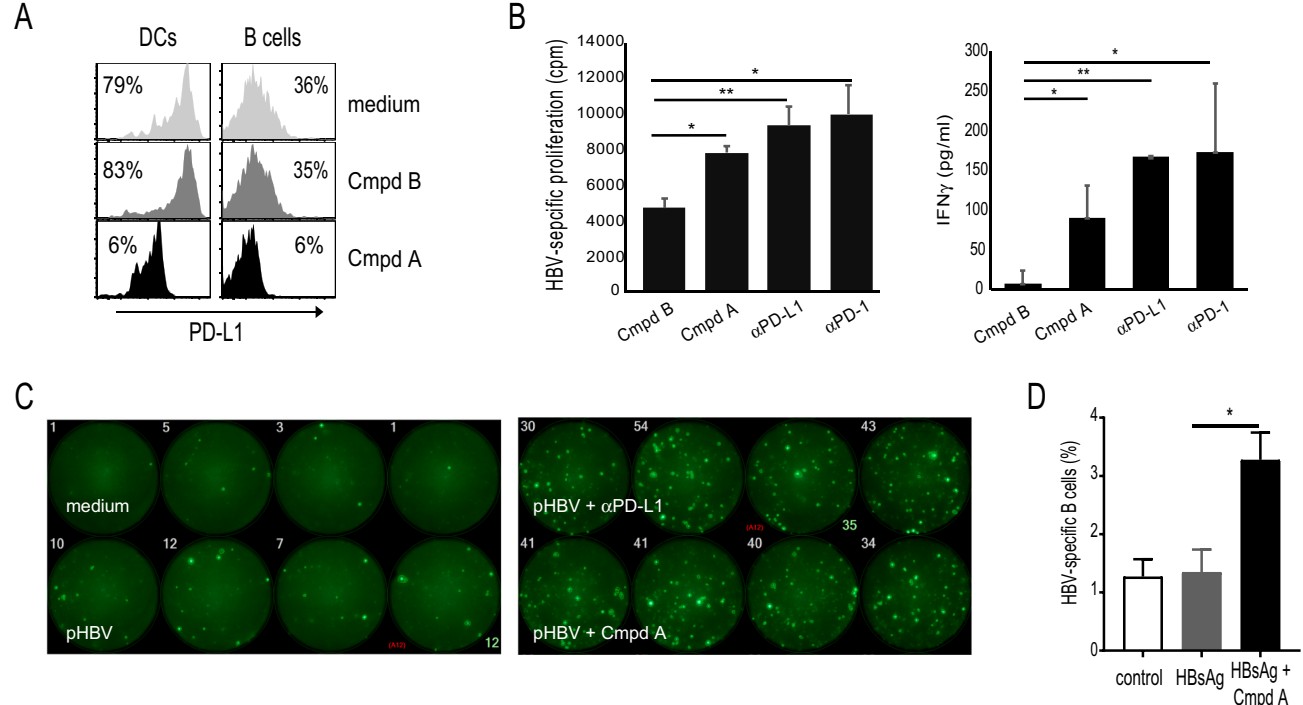

**Fig. 4 Effect of compound A on ex vivo HBV-specific immune responses. a** PBMCs stimulated with compound A for 1 h, followed by flow cytometric measurement of cell surface PD-L1 percentage. **b** PBMCs cultured in triplicate wells with HBV peptides for 10 days and T cell proliferation (thymidine incorporation) and IFNγ secretion (ELISA) measured in the presence of compounds and antibodies. **c** PBMCs stimulated with compounds or antibodies overnight followed by IFNγ producing cells measured by Fluorospot. **d** PBMCs collected from HBV vaccinated individuals ($n = 3$) on day 7 post booster vaccination were incubated with either rHBsAg alone (1 μg/ml) or rHBsAg plus compound A at 0.5 μM concentration for 5 days. Cells were then collected and stained for HBsAg-specific B cells or plasmablasts (PBs), using Atto488-labeled rHBsAg by flow cytometry. The error bars represent the standard error of mean of three individuals. statistical significance (*$p \le 0.05$, **$p \le 0.01$, ***$p \le 0.001$) was determined by unpaired two-tailed Student's $t$ test. Source data are provided as a Source data file.

treated with the αPD-L1 antibody reference atezolizumab, and are supportive of a PD-L1-based mechanism of action for compound A.

**Antiviral effects of small-molecule PD-L1 inhibition.** To test compound A effects on normal human immune cells, dendritic cells (DCs) and B cells were treated with compound A for 1 h and cell surface PD-L1 measured by flow cytometry (Fig. 4A). Compound A reduced cell surface PD-L1 in both DCs and B cells, suggesting broad primary cell-type effects on PD-L1 internalization (Fig. 4A). It has been shown that αPD-1/αPD-L1 antibodies can boost T and B cell responses to HBV antigens[31,44]. To extend these findings, to a small-molecule checkpoint inhibitor with a completely different mechanism of action, we treated PBMCs isolated from chronic HBV patients with HBV peptides, and tested compound A effects on proliferation and IFNγ T cell responses. As with atezolizumab (αPD-L1) and nivolumab (αPD-1), compound A treatment boosted HBV-specific T cell proliferative and cytokine responses to HBV antigens (Fig. 4B). Consistent with these results, treatment of HBV donor PBMCs with αPD-L1 antibody (MIH1) or compound A resulted in a significant increase in IFNγ-secreting T cells as measured by fluorospot (Fig. 4C). It has been shown previously that HBV-specific B cell responses are enhanced with nivolumab (αPD-1) treatment[31]. To test if compound A could enhance HBV-specific B cell responses, we stimulated HBV vaccinee PBMCs with recombinant HBsAg in the presence and absence of compound. Induction of HBsAg-specific B cells was minimal by rHBsAg stimulation alone; however, compound A increased the frequency of HBsAg-specific B cells ($p = 0.04$). Therefore, these data

demonstrate that compound A not only activated Ag-specific T cells (IFN-γ production), but also directly stimulated Ag-specific B cells to proliferate and differentiate into Ab-secreting cells. Taken together, these results indicate that treatment with compound A could enhance HBV-specific immune responses.

**Crystal structure of compound A in complex with PD-L1 dimer.** The structure of compound A is disclosed in Supplemental Fig. 2A. Compound A was co-crystallized with hPD-L1 in a dimeric form and the obtained crystals diffracted to the 2.49 Å resolution (Supplemental Fig. 2B). The comprehensive crystal structure data are highlighted in Supplementary Table 1. Compound A interacts with both protein subunits, burying 276.0 Å² of chain A and 271.0 Å² of chain B (Supplemental Fig. 2C). Both left-hand side (LHS) and right-hand side (RHS; ethylamino) ethanol form hydrogen bonding interaction with Asp122 of chain A and chain B, respectively. The LHS and RHS pyridine rings form π–π stacking interactions with Tyr56. The LHS amide linker carboxylate makes water-mediated hydrogen bond interactions to Gln66 of chain A and a similar interaction was observed on RHS to chain B. The central benzyls bind between Ala121 side chain of one chain and Met115 side chain of the other chain. These aryls also form van der Waals interactions with Asp122, Tyr123, Ile54, and Ser117. The two linked central benzyls are rotated by −106° with respect to each other at chain A:B interface (Supplemental Fig. 2C).

**Discussion**
Clinical administration of antibodies targeting the PD-1/PD-L1 interaction has been a successful immunotherapy in treating

various malignancies[9,12,13,45,46]. To date, drug discovery efforts to identify functionally similar small-molecule checkpoint inhibitors have proven unsuccessful. In this study, we report on a small molecule which like α-PD-L1/PD-1 antibodies can stimulate human adaptive immune responses. Here, we have shown that compound A inhibits the PD-1/PD-L1 axis by inducing cell surface PD-L1 dimerization followed by rapid internalization into the cytosol. PD-L1 is no longer available on the cell surface to interact with its cognate receptor (PD-1) and likely not available for other known cell surface interactions, such as CD80. This compound A-induced loss of cell surface PD-L1 inhibits in vivo tumor growth, and can enhance hepatitis B virus-specific T and B cell responses in patient samples.

A previous report identified small molecules which bind and induce dimer formation of recombinant PD-L1 protein in solution; however, it was not reported if this observation was simply an artifact of biochemical systems or could also be observed in vivo[23,47]. Using a cell-based NFAT reporter system (Jurkat-CHO-PD-1/PD-L1 bioassay), we observed that compound A is a potent (17 nM EC$_{50}$) inhibitor of PD-1/PD-L1 in living cells (Fig. 1C). It is known that αPD-1/PD-L1 antibodies inhibit this checkpoint axis through direct binding of the cognizant target protein in a region that includes the ligand interaction site, thereby disrupting the PD-1/PD-L1 intermolecular interaction[48,49]. Crystallography studies revealed that compound A can indeed bind PD-L1, but through a hydrophobic pocket, which is created between two PD-L1 molecules thereby stabilizing a PD-L1 homodimer. This compound A-induced dimer interface includes the residues needed to interact with PD-1 protein, thereby resulting in potent PD-1/PD-L1 HTRF (Fig. 1A and Supplemental Fig. 2B). It was evident from this 2.49 Å resolution crystal structure that the perpendicular bi-phenyl bond angle allowing for peripheral π–π stacking with antiparallel tyrosine residues is critical for compound A binding mode (Supplemental Fig. 2C). This was consistent with the binding mode observed in previous cell-free biochemical studies performed for similar chemotypes[23]. Based on this preference for compound A to form dimers in our crystallography studies[23,47], we theorized that compound A may function through a mechanism of action distinct from αPD-L1 antibodies. Interestingly, we found that these compounds induce a rapid internalization (<10 min) of cell surface PD-L1 (Supplemental Fig. 2C). While there are ample reports of other receptor and/or ligand internalizations, this is the first report of compound-induced PD-L1 internalization.

When measuring cell surface PD-L1 by flow cytometry in the absence of compound A, we see that constitutive cell surface occupancy levels (MFI) are relatively unchanged over time. However, this does not rule out the possibility that cell surface PD-L1 is being rapidly turned over with cytosolic homeostatic mechanisms capable of regulating stable cell surface levels. Therefore, it is conceivable that compound A could simply be inducing an imbalance of cell surface PD-L1 occupancy through disruption of the natural PD-L1 cell surface recycling process. This does not appear to be the case, since the internalization we see here is quite rapid (<10 min) and is accompanied by an induced change in PD-L1 intermolecular interactions (dimerization; Figs. 1B and 2A). In addition, consistent with the known effects of temperature on endocytosis, compound A-induced internalization can be significantly inhibited upon cellular exposure to 4 °C when compared to 37 °C. From these data, it appears that compound A is functioning by stimulating PD-L1 homodimeric interactions, which triggers uptake into the cytosol. Future studies will be focused on the precise mechanism of this two-step internalization process. Based on these findings our current working model for compound A small-molecule checkpoint inhibition is that upon PD-L1 binding of compound A, cis-interacting homodimers are formed, triggering a rapid loss of cell surface PD-L1 through internalization into the cytosol, thereby preventing any further interaction with PD-1-expressing cell types. Elucidation of the crystal structure indeed illustrated the formation of a homodimer with compound A bound within a hydrophobic pocket created between two PD-L1 proteins. This pocket is comprised of antiparallel β-sheets and mirror image π-stacking tyrosine residues, which do not require 180° rotation of the full-length PD-L1 protein. We believe these data suggest that PD-L1 can come together on the plasma membrane in a cis-interacting conformation, but still forming the antiparallel and symmetrical compound pocket. In addition, we see dimerization of recombinant PD-L1 when treated with compound A, suggesting the dimerizing effect of compound A on PD-L1 is direct and does not require other host proteins. We also see that this cell surface inhibition of PD-L1 can be sustained if exposure to compound A remains elevated. Interestingly, upon complete compound A removal, we see that cell surface PD-L1 is fully reconstituted within 48 h, which requires newly transcribed PD-L1 mRNA and golgi-mediated protein transport (Fig. 2F, G). These data suggest the compound A-induced internalization mechanism is fully reversible and that the previously internalized PD-L1 is not simply recycled back to the cell membrane. It is certainly possible that this small-molecule loss of cell surface PD-L1 as observed with compound A treatment could result in bonafide in vivo immunomodulatory activity, which we confirmed in a humanized murine model for colon cancer and in virally infected patient samples.

In order to extend our in vitro findings to antigens known to induce immune tolerance through checkpoint upregulation, we first tested if treatment with compound A could upregulate human PBMC responses to CMV. It is known that αPD-1/PD-L1 antibodies can break tolerance to CMV antigens when tested on ex vivo PBMCs isolated from CMV-positive human subjects[48]. Interestingly, identical to the tested αPD-L1 antibody, compound A is a potent activator of human CMV recall responses (Fig. 1D). These data suggest that while compound A inhibits the PD-1/PD-L1 immune checkpoint axis in a different manner than αPD-1 or αPD-L1 antibodies, this mechanism of action can result in immunomodulation in keeping with that seen for these antibody therapies. To further extend these findings to an in vivo model, we tested compound A in a mouse model for colorectal cancer. In vivo efficacy assessment of checkpoint inhibitors has historically centered around demonstration of antitumor effects in syngeneic or tumor xenograft mouse models. As compound A binds specifically to only human PD-L1, we conducted efficacy evaluation of compound A in a mouse colorectal cancer model where both host and tumor are engineered to express human PD-L1. Compound A treatment resulted in the significant tumor growth inhibition with off-treatment tumor stasis observed in three out of eight animals, and the pretreatment tumor burden completely abolished (full regression) in one other animal (Fig. 3A). Immunophenotyping of circulating T cell populations in response to compound A treatment indicated increases in CD3$^+$ T cell numbers, of which CD4$^+$ T cells were significantly elevated, as well as significant decreases in T regulatory cell frequency (Fig. 3C, D). These changes mediated by compound A paralleled immune signatures observed with atezolizumab treatment and are supportive of a PD-L1-based mechanism of action. It is striking that, in four out of eight animals, only 7 days of compound A treatment of mice with established tumors could result in a similar degree of tumor growth inhibition as animals administered atezolizumab for 16 days. Future studies will be

focused on expanding these findings to other tumor types and evaluations of optimal treatment duration and dose.

The role of PD-1/PD-L1 immune checkpoint in the development and persistence of chronic viral infections has been well established[50]. One such virus with significant implications to world health is chronic hepatitis B (CHB) infection. A hallmark in the development and persistence of chronic HBV is thought to include PD-1/PD-L1-mediated induction of immune tolerance to HBV antigens. Consistent with the parallel activities of compound A and αPD-L1 in the mouse oncology model, we also observed comparable immunomodulatory effects on human ex vivo HBV-specific T and B cell responses. It is important to note that the compound A-induced loss of cell surface PD-L1 is not cell-type specific, as we see these effects on both DCs and B cells (Fig. 4A). In addition, this is the first report of a small-molecule checkpoint inhibitor inducing T cell proliferative and IFNγ responses to HBV peptides (Fig. 4B, C). While it is not yet clear what level of patient HBV immune reactivation might tip the balance toward full immune reconstitution and clearance of HBV infection, the identification of compound A-mediated checkpoint inhibition could be an avenue for pursuit of the development of curative CHB immune therapies. Also, the identification of a potentially more tolerable and reversible small-molecule approach to checkpoint modulation could be particularly impactful for chronic viral infections, such as CHB, where many patients are asymptomatic and the known side effects associated with antibody therapy are not acceptable for this patient population. This study has identified a previously unknown mechanism of PD-1/PD-L1 checkpoint inhibition through small-molecule induction of PD-L1 internalization, which could have a significant impact in advancing our understanding of checkpoint biology and therapeutics.

## Methods

**Ethic statement**. This study regarding human blood samples was approved by the institutional review boards of Quorum Review (USA) and Western IRB (USA), and informed consent was obtained from each patient. This study with mice was conducted by Biocytogen in accordance with AAALAC guidelines. This study was approved by Biocytogen's Institutional Animal Care and Use Committee and in compliance with the Guide for the Care and Use of Laboratory Animals (National Research Council, 2011).

**Human blood samples**. All human blood samples were purchased from Sanguine (CA, USA) and BioIVT (NY, USA). Human samples were approved by the Institutional review boards of Quorum Review (Seattle, WA) and Western (USA), and informed consent was obtained from each patient.

**Mice**. Female double knock-in mice for humanized PD-1 and PD-L1 (C57BL/6-Pdcd1[tm1(PDCD1)] Cd274[tm1(CD274)/Bcgen]) were implanted with MC38 cells expressing humanized PD-L1 (MC38 Cd274[tm1(CAG-CD274)Bcgen]) subcutaneously in the right flank ($1 \times 10^6$ cells per mouse). Twelve days later, when tumor volumes reached ~100 mm$^3$ (75–125 mm$^3$) as measured by digital caliper, animals were randomized based on tumor volumes into treatment groups ($n = 8$ per group). Compound A treatment was administered via oral gavage at 10 mg/kg for 7 daily doses between study days 0 and 6. Atezolizumab was administered via intraperitoneal injection at 10 mg/kg thrice weekly between study days 0 and 16. Upon study termination at study day 28, blood and tumor samples were assessed by flow cytometry analysis for the markers live/dead (catalog no. 423101, BioLegend), CD45 (catalog no. 103154, BioLegend), CD3 (catalog no. 100210, BioLegend), CD8 (catalog no. 100724, BioLegend), CD4 (catalog no. 100453, BioLegend), CD25 (catalog no. 102030, BioLegend), FoxP3 (catalog no. 12-5773−80, Thermo Scientific), and human PD-L1 (catalog no. 329724, BioLegend). Tumor concentrations of compound A were assessed by LC-MS/MS (Sciex Qtrap 5500 system).

**Homogeneous time-resolved fluorescence assay**. Recombinant Human PD-L1-His (R&D systems) at 6 nM, small molecules and antibodies were added to 384-well plate (Corning) followed by recombinant Human PD-1 Fc Chimera (R&D systems) at 6 nM. pAb anti-Human IgG-XL665 (Cisbio) and mAb anti-6His Tb cryptate Gold (Cisbio) were added to the 384-well plate after mixing at 6.7 and 0.35 nM concentration, respectively. For dimerization assay PD-L1 Fc was used instead of PD-1 Fc, and incubated in total volume of 20 µl per well overnight at room temperature and read by Envision (PerkinElmer, MA).

**BioAssay**. BioAssay is designed to measure luciferase activity showing the $EC_{50}$ level of PD-1/PD-L1 blocking by small molecules targeting PD-L1 protein in the cell. PD-1 effector cells were incubated with PD-L1 aAPC/CHO-K1 cells in the absence or presence of small molecules or αPD-L1 blocking antibodies, as indicated titration according to the manufacturer's instruction (Promega). Bio-Glo™ Reagent was added and luminescence quantified. Data were analyzed using GraphPad Prism.

**Native gel**. PD-L1 aAPC/CHO-K1 cells were incubated overnight at 1 million per well in a six-well plate in a 37 °C CO$_2$ incubator. Cells were treated the next day with 1 µM small molecules for 1 h and extracted using M-PER™ Mammalian Protein Extraction Reagent (ThermoFisher) plus 10% DDM on an agitator at 4 °C for 1 h. The lysates were centrifuged at $14,000 \times g$ for 15 min at 4 °C, and 2 µL supernatants were treated with 1 µL of G-250 dye and 5 µL 4× Native Page sample buffer (ThermoFisher). Samples and Native Mark ladder were run on 4–16% Native Page gel (ThermoFisher) at 150 V constant for 1 h and 55 min with Anode Buffer in outer chamber and Cathode Buffer (light blue 0.002% Coomassie) in inner chamber (ThermoFisher). Membrane transfer was done using Trans-Blot Turbo (BioRad) and PVDF membrane mini transfer packs (BioRad). PVDF membrane is presoaked in methanol before transfer to facilitate transfer of Native Page ladder. Using Pierce™ Reversible stain kit (ThemoFisher), PVDF membrane was stained with Memcode for 30 min. Stain was reversed over the PD-L1 bands by washing part of the membrane that does not contain the Native Page ladder. The protein ladder was distained by washing until bands become visible. Membrane was blocked with 5% milk and TBS for 1 h at room temperature with gentle agitation. Primary antibody used was polyclonal αPD-L1 goat IgG antibody (R&D systems Cat No. AF156) at 2 µg/ml in 5% milk and TBS and incubated overnight at 4 °C with agitation. PVDF membrane was washed three times with TBS-T the next day and incubated with secondary antibody donkey anti-goat (Millipore Cat No. AP180P) at 1:10,000 in 5% milk and TBS-T for 1 h at room temperature with gentle agitation. PVDF membrane was washed three times with TBS-T and immunoblots developed with SuperSignal West PicoPLUS substrate (ThermoFisher).

**Cell culture and transfection**. PD-L1-negative cells (aAPC/CHO-K1, Promega) are CHO-K1 cells with an engineered cell surface protein designed to activate cognate TCRs in an antigen-independent manner. Cells were maintained in Ham's F-12 Nutrient Mix media (Gibco) supplemented with 10% fetal bovine serum (FBS) and 1% penicillin–streptomycin. Transient transfections were performed with Lipofectamine™ 3000 Transfection Reagent (Invitrogen) according to manufactures' instructions.

**Plasmids**. Human PD-L1 was cloned into pcDNA3.1(+) expression vector. Flag- and cMyc-tagged human PD-L1 were generated by introducing the tags at the C-terminus (cytoplasmatic) and N-terminus (extracellular) domains, respectively. Thus, Flag was expressed in cytosol, whereas cMyc was expressed outside of cell membrane. All constructs were confirmed by using enzyme digestion and sequencing analysis (GenScript, NJ).

**Immunoprecipitation and Immunoblot analysis**. Transfected aAPC/CHO-K1 cells with plasmids were lysed in Pierce™ IP Lysis Buffer (25 mM Tris·HCl pH7.4, 150 mM NaCl, 1% NP-40, 1 mM EDTA, and 5% glycerol, Thermo Scientific) supplemented with Halt™ Protease Inhibitor Cocktail and Halt™ Phosphatase Inhibitor Cocktail (Thermo Scientific). Anti-cMyc immunoprecipitation was performed using Pierce™ anti-cMyc Magnetic Beads (Thermo Scientific). For co-expression of PD-L1-cMyc and PD-L1-Flag, $2 \times 10^6$ aAPC/CHO-K1 cells were plated in a 100 mm TC-treated culture dish overnight at 37 °C in 5% CO$_2$ conditions. Cells were transfected with 4 µg of each DNA plasmid. For single transfections, we used 8 µg of DNA plasmid. For immunoblotting, proteins were resolved by NuPAGE™ 4–12% Bis-Tris Protein Gels (SDS–polyacrylamide gel electrophoresis, Invitrogen, Cat No. NP0329BOX) and transferred onto a PVDF membrane, using Trans-Blot® Turbo™ Mini PVDF Transfer Pack (BioRad, Cat No. 1704156). The following antibodies were used: anti-Flag (DDDDK) tag antibody (Clone M2)-HRP (Abcam, Cat No. ab49763, 1:1000), anti-Myc antibody (Clone 9E10)-HRP (Abcam, Cat No. ab62928, 1:10000), and β-actin monoclonal antibody (Clone BA3R)-HRP (Invitrogen, Cat No. MA5-15739-HRP, 1:1000). Immunoblots were developed with SuperSignal™ West Femto Maximum Sensitivity Substrate (Thermo Scientific).

**Live cell imaging**. A total of $2 \times 10^5$ aAPC/CHO-K1 cells were seeded into Nunc™ Glass bottom 35 mm dishes (Thermo Scientific) and incubated overnight at 37 °C in 5% CO$_2$ conditions. The next day cells were transfected with the plasmid harboring PD-L1-cMyc, as previously described. Next day, cells were washed by PBS and incubated for 30 min at room temperature with anti-cMyc monoclonal antibody (Clone 9E10) Alexa Fluor® 488 (Invitrogen) at 1:1000 dilution in FluoroBrite™ DMEM media (Gibco) supplemented with 10% FBS and GlutaMAX™ supplement (Gibco). Nuclear staining was achieved using NucBlue™ Live ReadyProbes™ Reagent (Invitrogen) by manufacturer's instructions. Cells were washed with media twice and placed in an LCI Chamlide™ stage top

incubator system at 37 °C in 5% $CO_2$ conditions. Live cell imaging of labeled proteins was performed using an Olympus IX71 inverted microscope Visitech VT iSIM scan head for real-time 2× spatial resolution enhancement in x, y, and z. Images were acquired with a 100×/1.40 NA Oil objective and detected by a Hamamatsu ORCA-Flash4.0 Digital sCMOS camera. To obtain high-resolution three-dimensional images of the cells, Z-stacks were obtained using MetaMorph software (Molecular Devices, Sunnyvale, CA), and then the images were imported into Fiji (ImageJ 1.52n) software for analysis[51,52]. Typically, 10–15 serial two-dimensional confocal images were recorded at 25 nm intervals. All image acquisition settings were identical for all experimental variants in each experiment.

**Quantitative assessment by LC-MS/MS**. Cell lysate (50 µl) was treated with three volumes of acetonitrile containing internal standard (100 ng/ml tolbutamide) to precipitate proteins. After vortexing for 5 min, the mixture was centrifuged at 1643 × g at 4 °C for 10 min. The supernatant was then injected onto the LC-MS/MS system for analysis. The LC-MS/MS system consisted of a QTRAP 5500 mass spectrometer with electrospray ion source (ESI; Sciex, Redwood city, CA) equipped with a Shimadzu Nexera XR UPLC system (Columbia, MD). Compound A and internal standard were eluted from an Acquity UPLC HSS T3 column (2.1 × 50 mm, 1.8 µm particle size, Waters, Milford, MA) using a gradient of 0.025% formic acid in water (mobile phase A) and 0.025% formic acid in 50:50 (v:v) acetonitrile:methanol (mobile phase B). Compound A and internal standard were detected in positive ESI mode by multiple reaction monitoring transitions of m/z 569.2 → 373.3 and 271.1 → 155.3, respectively. The concentration of compound A in study samples were calculated using the peak area ratios of unknown against a standard curve relating the peak area ratios to the analyte concentration in a linear, $1/x^2$ regression.

**CMV recall assay**. Cryopreserved CMV-seropositive PBMCs were thawed and rested overnight in AIM-V media (ThermoFisher). Cells were then stimulated with grade 2, CMV lysate (Microbix) at 0.5µg/ml, and CD28 and CD49 at 1 µg/ml with threefold serial dilution of appropriate controls and compound A, in U-bottomed 96-well plates. αPD-L1 antibody (eBioscience, clone MIH1) and mouse IgG1 kappa isotype control antibody (eBioscience), starting at 1µg/ml, were used as positive and negative controls, respectively. After incubation of cells for 48 h, culture supernatants were collected for analysis of IFNγ by ELISA. Briefly, IFNγ ELISA was performed in 96-well MaxiSorp ELISA plate 4HBX, coated with anti-IFNγ antibody (BD Bioscience) for overnight at 4 °C and blocked for 1 h with 3% BSA-containing PBS-Tween (0.05%). Cell culture supernatant and rIFNγ (BD Bioscience) as a control were incubated in ELISA plate for 2 h at room temperature and detected by biotinylated anti-IFNγ antibody (554550, BD Bioscience) followed by streptavidin–HRP (Southern Biotech). Signals were then developed by TMB (eBioscience) and stop solution (0.05 M HCl or $H_2SO_4$).

**PBMC internalization assay**. Cryopreserved CMV-seropositive PBMCs were revitalized and stimulated with CMV lysate for 24 h. Cells were then incubated with compounds at 1 µM or with IgG at 1 µg/ml for additional 24 h. Aliquots of cells were collected and stained for surface PD-L1 expression assessment by flow cytometry at 4 °C at designated timepoints up to 24 h. Briefly, cells collected at designated timepoints were washed and stained with APC-conjugated αPD-L1 antibody on B cells (CD20+) and DCs (CD11c+). In addition, PD-1 expression was assessed on T cells (CD4 and CD8) with PE-conjugated αPD-1 antibody. Dead cells were excluded using Live/Dead™ fixable Aqua dead cell stain kit (ThermoFisher). Cells were acquired by LSRFortessa (BD Bioscience, CA) and analyzed by FlowJo (TreeStar, OR).

**Acid wash internalization assay**. PD-L1 aAPC/CHO-K1 cells were incubated with small molecules or antibodies (purified αPD-L1 antibody clone MIH1, or Opdivo αPD-1 antibody) for 1 h and washed with 1× PBS three times. Some cells were further washed with acid wash buffer (DMEM, 0.2 % BSA, pH 3.5) three times for 5 min on the shaker to strip off the small molecules and antibody bound to PD-L1 on the cell surface, then another three washes with 1× PBS. Lift buffer (10 mM Tris, 140 mM) was used to detach the cells from the cell plate. Cells treated with or without acid wash were stained with αPD-L1 antibody (clone MIH1, Invitrogen) and Live/Dead™ fixable Aqua dead cell stain kit (L34965, ThermoFisher). Cells were acquired by LSRFortessa (BD Bioscience, CA) and analyzed by Flowjo (TreeStar, OR).

**B cell in vitro culture**. Cryopreserved PBMCs were thawed and rested overnight in RPMI-1640 media supplemented with 10% FBS, pen/strep, HEPES, and beta-mercaptoethanol. Cells were then stimulated with rHBsAg (subtype ayw, adw, adr; Fitzgerald 30R-AH018, 30R-AH016, 30—AH37) at 1 µg/ml in the absence or presence of compound A (0.5 µM) for 5 days in U-bottom 96-well plate. Cells were collected and stained for HBsAg-specific B cells and plasmablasts (Ab-secreting cells) by flow cytometry. Briefly, cells were washed and stained with rHBsAg

labeled with Atto488 (Innova Bioscience), APC-Cy7-anti-CD20, Alexa700-anti-CD38, Brilliant Violet 785-anti-CD27, Brilliant Violet 650-anti-CD3, and Brilliant Violet 605-anti-IgD. The frequencies of HBsAg-specific B cells were analyzed from surface staining of Atto488-HBsAg-binding memory B cells (CD3−CD20+IgD−CD27+). The frequencies of HBsAg-specific plasmablasts were analyzed from intracellular staining of Atto488-HBsAg of plasmablasts (CD3− CD20−CD38+CD27+).

**Thymidine proliferation assay**. Details of the thymidine proliferation assay have been previously described[44,53]. Briefly, PBMCs from HBV donors were isolated and were cultured with RPMI containing 2% human AB serum with pen/strep for 14 days in the presence of medium, HBV overlapping peptides of S or Core, influenza overlapping peptides, as well as small molecules (0.5 µM), or antibodies (5 µg/µl), including atezolizumab and nivolumab[44]. HBV genotype-specific over-lapping 15-mer peptides (genotypes A–D) were synthesized by GenScript (NJ, USA). A total of 1 µCi of $^3$H thymidine (NET355001MC, PerkinElmer) was added for 18 h for its incorporation into newly synthesized chromosomal DNA before harvesting cells. $^3$H thymidine incorporation was counted by MicroBeta2 micro-plate scintillation counter (PerkinElmer, MA). A total of 50 µl of supernatants per well before adding $^3$H thymidine was used for measuring IFNγ production by ELISA (430107, BioLegend).

**Cold internalization assay**. PD-L1 expressing CHO cells were placed in 4 °C or 37 °C for 30 min followed by the treatment with media or 100 nM compound A. Cells were incubated with compound A for 30 min followed by cell surface PD-L1 quantitation by flow cytometry, as described previously (MFI%; * denotes $p < 0.05$).

**Co-crystal structure hPD-L1 with compound A**. The DNA coding for the human PD-L1(18–134) with 6xHis carboxyl-terminal tag was cloned in pET28 vector and the intended recombinant protein overexpressed in Escherichia coli BL21 (DE3). The purified inclusion bodies were solubilized in 50 mM Tris HCl, pH 8.0, 200 mM NaCl, 10 mM 2-mercaptoethanol, 6 M guanidium HCl, and the protein was refolded by dilution with 100 mM Tris HCl, pH 8.0, 1 M arginine, 0.25 mM reduced-GSH, and 0.25 mM oxidized-GSH. The refolded protein was purified by Superdex S75 equilibrated with 10 mM Tris HCl, pH 8.0 and 20 mM NaCl. hPD-L1 protein at 0.6 mg/ml with 180-fold molar excess compound A mixed in was incubated on ice for 60 min. The resulting protein complex was concentrated to 6 mg/ml and screened for crystallization at 20 °C using 1:1 ratio protein:reservoir, using the sitting drop vapor diffusion method. Orthorhombic monocrystals appeared in 8–13 days, where the reservoir formulation contained 100 mM NaCitrate, pH 6.5 and 22% (w/v) PEG 3000. The harvested crystals were treated with 20% (v/v) 2-methyl-2,4-pentanediol and flash-frozen in liquid nitrogen. X-ray diffraction data were measured using Pilatus 6 M detector and a wavelength of 0.97928 Å at the XALOC beamline, ALBA, Barcelona, Spain. The data were indexed and integrated using XDS and scaled and merged using Aimless. The structure was solved by molecular replacement using MOLREP and 4ZQK as search model. The model was rebuilt using Coot and refined with REFMAC5. The structure has been deposited with the RCSB as entry codes 6VQN.pdb.

**Statistical analysis**. Data are expressed as mean ± standard deviation. Unpaired or paired Student's t test was used for comparison of the two groups. The correlation among Internalization, HTRF, and BioAssay was evaluated by Pearson's correlation test. Statistically significant differences of $p \le 0.05$, $p \le 0.01$, and $p \le 0.001$ are noted with *, **, and ***, respectively.

**Reporting summary**. Further information on research design is available in the Nature Research Reporting Summary linked to this article.

## Data availability
Crystal structure data that support the findings of this study have been deposited in RCSB protein database (6VQN.pdb). The authors declare that all other data supporting the findings of this study are available within the paper and its Supplementary Information files. Source data are provided with this paper.

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

## Acknowledgements

We thank Dr. Andrea L. Stout for the technical assistance in study using confocal microscopy at CDB Microscopy Core Facility, University of Pennsylvania. We thank all the study subjects for their participation.

## Author contributions

C.B.M. conceived and designed experiments and wrote manuscript; J.-J.P., V.H.C., C.L.I., J.H.K., and B.S.P. performed biological in vitro and ex vivo experiments; E.P.T., A.C.H.L., and S.K. designed and conducted mouse tumor studies; Y.B., D.N., A.G.C., and B.D.D. designed and synthesized molecules; K.F. performed computational chemistry; T.H., R.Q., and S.T. performed pharmacokinetic analysis; A.W., D.L., C.L., and R.K.S. designed and performed crystallographic experiments; R.R. designed biological experiments and revised manuscript; and M.S. provided project oversight, revised manuscript, and designed molecules.

## Competing interests

The authors declare no competing interests.
