## [Peer Review File · Nature Communications]

Reviewers' comments:

Reviewer #1 (Remarks to the Author):

In this brief manuscript, Park et al. identified a novel small molecule compound (CmpdA) which induces PD-L1 dimerization in solution, inhibits PD-L1/PD-1 signalling in vitro, inhibits tumor growth in a humanized mouse colorectal model, and the HBV-specific responses ex vivo. In the latter two scenarios, the efficacy of CmpdA appeared to rival PD-1 and PD-L1 inhibitors used in clinics. They also found that CmpdA causes PD-L1 internalization, thereby reducing its cell surface expression. They proposed that internalization of PD-L1 is the major mechanism by which CmpdA restores T cell functions, although they also have data suggesting that CmpdA inhibits PD-L1/PD-1 interaction. Overall this is an interesting study with implications in immunotherapies against cancer and chronic viral infections, but I have the following concerns.

1. The authors presented conflicting evidence/statements on whether Cmpd A inhibits PD-L1/PD-1 interaction. They stated that "Interestingly, we found that rather than directly blocking the PD-1/PD-L1 interaction, these compounds induce a rapid internalization (<10mins) of cell surface PD-L1", but it was clear from Fig. 1A that Cmpd A inhibits PD-1/PD-L1 binding at least in solution. I feel that the authors could test how CmpdA affects the staining of PD-1-Fc to PD-L1 expressing cells. This experiment is very easy to do. The potential endocytosis of PD-L1 can be inhibited either by drugs or low temperature.
2. While the HTRF assay suggests that Cmpd A induces dimerization of PD-L1, there is no data to show that it can do so in cell culture assays. Fig. 1C that Cmpd A inhibits PD-L1 signaling in the reporter Jurkat cells, but this assay does not prove that Cmpd A dimerizes PD-L1 in cells. If the two PD-L1 molecules crosslinked by Cmpd A are indeed antiparallel, it is difficult to imagine how dimerization could occur on cell membranes, in which PD-L1 are anchored in the same orientation.
3. Line 138, the authors stated that "Interestingly, compound A induction of PD-L1 dimerization requires living cells", based on their cell lysate experiments. If so, then why did they detect Cmpd A induced dimerization in Fig. 1B?
4. Fig. 1B, only one concentration tested, I'd like to see a dose response.
5. Fig. 1C did not rigorously prove that Cmpd A inhibits PD-L1/PD-1 signaling, because they did not exclude the possibility that Cmpd A works through other signaling pathways or some pleiotropic mechanisms. For example, hydrophobic compounds may induce cell clustering, which is not evaluated in the current manuscript. I also feel that a PD-1 KO or PD-L1 KO condition is required. If the authors' model is correct, then they should detect no effect of Cmpd A under either condition. Alternatively, they could test the effect of Cmpd A in the presence of saturating concentrations of anti-PD-L1 or anti-PD-1.
6. It was recently shown that PD-L1 interact with CD80 in cis to affect both PD-1 and CTLA-4 pathways. If Cmpd A induces PD-L1 homodimerization, how does it affect PD-L1/CD80 interaction, and how would this impact their interpretation of their results?
7. The manuscript is concisely written, but the authors can better describe the methods and materials. As it stands, there is not enough information on how each experiment was conducted. For example line 411, why using anti-His6?

Reviewer #2 (Remarks to the Author):

In the manuscript, Park and co-workers present the discovery and biological activity of a symmetric small molecule inhibitor of PD-1/PD-L1 interaction. The authors also propose a unique mechanism of action of this inhibitor, which acts by inducing dimerization and internalization of PD-L1 in model CHO cells.

The manuscript is well written and the results are presented in an easy-to-follow form. The findings provide interesting mechanism of action of the molecule and contribute well to the progress in the field of targeting PD-L1 with small molecules. The reviewer suggests considering the publication of the manuscript, provided that the authors address the following (minor) questions/comments, which include lack of several critical controls in some of the experiments.

Addressing these concerns would greatly increase the relevance of the conclusions drawn by the authors.

1. Figure 1C – is it truly a % inhibition? How many repeats were done?
2. Figure 2A,B – how is the PD-L1 detected in whole cell lysates prepared with the M-PER reagent?
3. Figure 2C, Figure 4A – flow cytometry: please clearly indicate the clones of antibodies that were used in the experiment. From the previous and following sections it can be assumed that MIH1 was used as an anti-PD-L1 control, and 29E.2A3 might have been used for FC. If this was the case, please analyze the possibility that the binding surface of 29E.2A3 overlaps with the binding surface of compound A and the second PD-L1 monomer. This would limit the binding of 29E.2A3 to PD-L1 in the presence of compound A, resulting in a lower FC signal.
4. Figure 2D – what does the phrase “Images are representative of three experiments” mean? How many individual cells were visualized in each experiment? Figures 2 F and G seemingly show quantified data from a similar experiment. How many cells were monitored in this experiment? Error bars are extremely low (G) or absent (F) – was the data reproducibility really so high? Please explain.
5. Figure 2E: the Co-IP experiment lacks necessary controls: the detection of Myc in precipitates is missing (the control of equal cMyc-PD-L1 amounts in DMSO and CmpdA eluates) and detection of Flag in input samples (the control of equal PD-L1-Flag amounts in DMSO- and CmpdA-treated cell lysates).
6. Figure 4B: why were the compound B-treated cells considered a control for the experiment and not the untreated cells?
7. Lines 263-264: “We believe this is the first report of a cellular potent low molecular weight small molecule PD-1/PD-L1 checkpoint inhibitor” – please refer to the previous manuscripts (Basu et. al 2019, DOI: 10.1021/acs.jmedchem.9b00795, and Skalniak et. al 2017, DOI: 10.18632/oncotarget.20050), where bioactive small molecules targeting PD-L1 were characterized.

Reviewer #3 (Remarks to the Author):

In the present study Park et al. describe the properties of a novel compound (named compound A) that induced internalization of PD-L1 resulting in suppression of PD-1:PD-L1 interaction, and its implications in T cell activation and anti-tumor function. The authors propose that use of this compound would have equivalent effects with antibodies blocking that PD-1 pathway and could potentially substitute for the use of such antibodies for clinical applications. Although the data are of potential interest several points require further investigation before conclusions can be made. Major points:

- 1) Important details on the HTRF assay developed by the authors are missing. This should be described in a comprehensive manner and the interpretation of the relevant results using this approach should be thoroughly outlined.
- 2) Compound A blocks PD-L1 interaction with PD-1. Does it also block PD-L1 interaction with B7-1?
- 3) The authors stated that ligand induced dimerization has been reported for surface receptors and use this as a justification to study the effects of compound A on PD-L1 dimerization. Although the nature of compound A is not disclosed, do they imply that compound A is a natural PD-L1 ligand that might mediate ligand induced dimerization? Without providing information whether compound A is a natural partner of PD-L1 this justification is scientifically inaccurate.
- 4) The authors support that compound A induced PD-L1 dimerization and subsequent internalization. There are no appropriate experimental data to support this conclusion. The assays shown using native gel electrophoresis do not provide evidence of the mechanism involved.

Specific methods are available to assess molecular dimerization at the cell membrane and such assays should be employed.

5) Figure 2A: The investigators used cell lysates to assess the effects of compound A on PD-L1 by native PAGE. The usual approach by which proteins are assessed after native PAGE is Coomassie staining. Is this what they did? There is no information how the authors assessed the proteins after electrophoresis. Cell lysates contain multiple proteins and the identity of the bands shown in the gels is unclear. Two separate bands are present at the area at which the authors indicate "PD-L1 monomer" but their identity is uncertain. CHO cells that do not express PD-L1 should also be used as control in this assay.

6) It is unclear whether compound A induces specifically PD-L1 dimerization or other cell surface proteins are involved in the observed effects. To clarify this, it is necessary to use purified PD-L1 protein to assess whether compound A can induce dimerization.

7) Figure 2C: The authors claim that treatment with compound A for 1 hour results in dimerization and internalization of PD-L1 leading to loss of surface PD-L1 expression as assessed by flow cytometry. It is necessary to perform detailed kinetics of PD-L1 expression levels on cell surface and cytoplasm by flow cytometry during multiple time points of treatment with compound A, to accurately assess changes on PD-L1 expression and subcellular localization.

8) Figure 2C: Which antibody clones were used for anti-PD-L1 staining? No specific information is provided. The PD-L1 Ab Biolegend #329724 is clone 29E.2A3 PD-L1 Ab, which blocks both PD-1 and B7-1 interactions with PD-L1. Since compound A also blocks the PD-1-PD-L1 interaction, this antibody should not be used to detect PD-L1 because potentially compound A interferes with staining. It is necessary to use multiple anti-PD-L1 antibody clones to assess parallel surface and cytoplasmic expression of PD-L1 after treatment with compound A or control.

9) The authors claim that treatment with compound A results in dimerization and internalization of PD-L1 but no such effect was observed after treatment with anti-PD-L1 or anti-PD-1 antibodies. Antibodies are bivalent and can dimerize PD-L1 as well, so why anti-PD-L1 antibody incubation does not cause internalization and loss of PD-L1 surface expression?

10) Figure 3A: Tumor experiment. Compound A was administered for 7 days and outcomes on targeted populations were assessed several days later (on day 28). According to the results shown in Figure 2F, wash out of compound A resulted in re-expression of PD-L1 to baseline levels. Based on these data, it is obvious that after cessation of in vivo administration, decrease of compound A levels will result in gradually diminished efficacy and recovery of PD-L1 expression on the targeted cells. Figure 3B shows decreased levels of PD-L1 expression on tumor CD45+ cells on day 28. It is not feasible to interpret these results without information about the pharmacokinetics and the clearance of compound A after in vivo administration. Furthermore, PD-L1 expression on target populations should be assessed at multiple time points after in vivo treatment.

11) The effects of compound A on PD-L1 expression on tumor cells should also be examined.

12) After analyzing the numbers of CD3+, CD4+ and CD8+ cells circulating in the blood, the authors concluded that the effects of compound A on these cell populations were similar to those induced by anti-PD-L1 antibody. However, the data show that compound A had only a slight effect on CD4+ T cells and no effect on CD8+ or total CD3+ T cells. In contrast, anti-PD-L1 antibody treatment resulted in significant expansion of all these T cell populations. Thus, these conclusions are inconsistent with the experimental data.

Minor points:

1) Line 123: Anti-PD-1 antibody is mentioned here instead of anti-PDL1 that is shown in the figure.

Reviewer Comments and Author Revisions:

Reviewer #1

In this brief manuscript, Park et al. identified a novel small molecule compound (CmpdA) which induces PD-L1 dimerization in solution, inhibits PD-L1/PD-1 signalling in vitro, inhibits tumor growth a humanized mouse colorectal model, and the HBV-specific responses ex vivo. In the latter two scenarios, the efficacy of CmpdA appeared to rival PD-1 and PD-L1 inhibitors used in clinics. They also found that CmpdA causes PD-L1 internalization, thereby reducing its cell surface expression. They proposed that internalization of PD-L1 is the major mechanism by which CmpdA restores T cell functions, although they also have data suggesting that CmpdA inhibits PD-L1/PD-1 interaction. Overall this is an interesting study with implications in immunotherapies against cancer and chronic viral infections, but I have the following concerns.

The authors would like to thank reviewer #1 for their thoughtful review and kind words on this manuscript. Below are the point by point detailed revisions and responses to each issue raised by reviewer #1. In addition, we have notated for ease of reference each revision in the manuscript based on this reviewer's suggestions.

1. The authors presented conflicting evidence/statements on whether Cmpd A inhibits PD-L1/PD-1 interaction. They stated that "Interestingly, we found that rather than directly blocking the PD-1/PD-L1 interaction, these compounds induce a rapid internalization (<10mins) of cell surface PD-L1", but it was clear from Fig. 1A that Cmpd A inhibits PD-1/PD-L1 binding at least in solution. I feel that the authors could test how CmpdA affects the staining of PD-1-Fc to PD-L1 expressing cells. This experiment is very easy to do. The potential endocytosis of PD-L1 can be inhibited either by drugs or low temperature.

We agree with the reviewer that the results from the PD-1/PD-L1 HTRF assay (Fig 1A) could initially appear conflicting with our later statements about compound mediated PD-L1 internalization. Our crystallographic (Sup Fig 2B) and homodimer HTRF (PD-L1/PD-L1; Fig 1B) studies indicate that compound A can in fact induce PD-L1 dimerization in solution. Analysis of the known interacting residues between PD-1 and PD-L1 proteins indicates that this compound A-induced PD-L1 dimerization covers the PD-1 interaction site, thus resulting in potent (pM) activity in the PD-1/PD-L1 HTRF assay (Fig 1A). This is in direct contrast to the steric hindrance of a bulky antibody on the PD-1/PD-L1 interaction. Of course, this PD-L1 solution-based dimerization could just be a biochemical artifact, therefore we next sought to determine if compound A-induced PD-L1 dimerization could occur in living cells. Dimerization indeed occurs in cells as seen in our PD-L1 native gels (Fig 2A;

Sup 3B). This dimerization is followed by a rapid loss of cell surface PD-L1 in PD-L1 expressing cells (Fig 2C,2D), therefore the PD-1-Fc staining experiment would not allow for measurement of PD-1/PD-L1 interactions since there is no PD-L1 on the cell surface upon compound A treatment after only 5mins. To better address what is happening in Fig 1A, we have added clarifying language to the results and discussion sections.

(page 6 108-111), Compound A, but not aPD-L1 (atezolizumab) or aPD-1 (nivolumab) induced PD-L1 protein intermolecular interactions, thereby increasing the HTRF fluorescent signal indicating compound A can induce PD-L1 homodimeric interactions (Fig 1B). In addition, we were able to visualize this solution-based compound A-induced PD-L1 dimerization through crystallographic studies (Supplemental Fig 2B).

(page 8 139-140), We observed compound A-induced PD-L1 intermolecular interactions in solution (Fig 1B). Changed to: We observed compound A-induced PD-L1 homodimeric interactions in solution (Fig 1B).

(page 15 288-293), Crystallography studies revealed that compound A can indeed bind PD-L1 through a hydrophobic pocket which is created between two PD-L1 molecules thereby stabilizing a PD-L1 homodimer. This compound A-induced dimer interface includes the residues needed to interact with PD-1 protein, thereby resulting in potent inhibition of PD-1/PD-L1 HTRF activity (Fig 1A; Supplemental Figure 2B).

(page 16 301-303), Interestingly, we found that rather than directly blocking the PD-1/PD-L1 interaction, these compounds induce a rapid internalization (<10mins) of cell surface PD-L1 (Fig 2E). Changed to: Interestingly, we found that these compounds induce a rapid internalization (<10mins) of cell surface PD-L1 (Sup Fig 2C).

We thank the reviewer for the suggestion to test inhibition of compound A PD-L1 endocytosis through exposing cells to low temperature. We feel this was a great idea and have completed this experiment. Indeed, we were able to significantly reduce compound A-induced PD-L1 internalization in 4° versus 37°. These data have now been included in a supplemental figure (Sup Fig 3A) and we have included details and discussion of this experiment in the methods, results, and discussion sections as shown below

*(page 33 651-655) **Cold Internalization Assay** PD-L1 expressing CHO cells were placed in 4° or 37° for 30mins followed by treatment with media or 100nM compound A. Cells were incubated with compound A for 30 minutes followed by cell surface PD-L1 quantitation by flow cytometry (MFI%) (*denotes $p < 0.05$).*

(page 9, 171-174) In addition, we were able to significantly inhibit compound A-induced PD-L1 internalization following pre-treatment exposure of CHO-PD-L1 expressing cells to 4°C (Supplemental Figure 3A).

(page 17, 316-319) In addition, consistent with the known effects of temperature on endocytosis, compound A-induced internalization can be significantly inhibited upon cellular exposure to 4°C when compared to 37°C.

2. While the HTRF assay suggests that Cmpd A induces dimerization of PD-L1, there is no data to show that it can do so in cell culture assays.

To answer this, we treated PD-L1 expressing CHO cells with compound A followed by detection of PD-L1 protein by native gel electrophoresis. We see a doubling (Sup Fig 3B) of PD-L1 protein upon compound A treatment suggesting compound-induced dimerization can occur in living cells.

Changed text: (page 8, 143-145) Within 1hr, compound A induced a shift to a larger molecular weight, doubling the observed monomeric form (Supplemental Figure 3B).

Fig. 1C that Cmpd A inhibits PD-L1 signaling in the reporter Jurkat cells, but this assay does not prove that Cmpd A dimerizes PD-L1 in cells. If the two PD-L1 molecules crosslinked by Cmpd A are indeed antiparallel, it is difficult to imagine how dimerization could occur on cell membranes, in which PD-L1 are anchored in the same orientation.

We agree with the reviewer that the Jurkat bioassay does not prove dimerization of PD-L1. This assay was designed to test whether the PD-1/PD-L1 interaction could be disrupted within a cell-based system. It was found that compound A is a potent inhibitor of cell based PD-1/PD-L1 interactions and with the crystallographic studies (Sup Fig 2b), cell based dimer blots (Fig 2A), and PD-L1 homodimer HTRF studies (Fig 1B) performed in this study, we believe the data in this manuscript supports that compound A inhibits the Jurkat PD-1/PD-L1 bioassay through rapid dimerization and internalization of cell surface PD-L1.

On the reviewer's point of antiparallel cross-linking, this is an excellent point and we completely agree that it is difficult to conceive how two PD-L1 molecules could re-arrange in an antiparallel fashion on the cell membrane. In fact, upon completion of our crystallographic studies, we see that the formation of the antiparallel compound A hydrophobic pocket does not require an antiparallel rearrangement of the full-length PD-L1 protein (Sup Fig 2B). Rather, we see that compound A interacts with only a few antiparallel residues close to the PD-L1 N-terminus. We believe our data suggests that compound A can bring PD-L1 molecules together in a *cis*

arrangement on the cell surface through an interface that includes antiparallel β -sheets and π -stacking mirror imaged tyrosine residues (Sup Fig 2C). To be clearer, we have included clarifying language in the manuscript as outlined below.

(page 17, 327-334), Elucidation of the crystal structure indeed illustrated the formation of a homodimer with compound A bound within a hydrophobic pocket created between two PD-L1 proteins. This pocket is comprised of antiparallel β -sheets and mirror image π -stacking tyrosine residues which do not require 180° rotation of the full-length PD-L1 protein. PD-L1 can come together on the plasma membrane in a cis-interacting conformation but still forming the antiparallel and symmetrical compound pocket.

3. Line 138, the authors stated that “Interestingly, compound A induction of PD-L1 dimerization requires living cells”, based on their cell lysate experiments. If so, then why did they detect Cmpd A induced dimerization in Fig. 1B?

We thank the reviewer for pointing out this somewhat misleading statement. We have modified this statement as follows and believe it now represents better what we were trying to convey.

(page 8, 148-155) Changed text to: We saw no post-lysis dimerization suggesting that in living cells an intact plasma membrane is required for compound A-induced PD-L1 dimerization. We can only surmise that this is because unlike the buffer-based cell-free dimerization observed in Fig 1B, PD-L1 proteins need to be in close proximity on the plasma membrane to induce homodimer formation and detergent free lysates are simply too viscous to allow for post-lysis PD-L1 dimerization. Nonetheless, we believe these data indicate that compound A dimerization is indeed occurring on the cell membrane in living cells.

4. Fig. 1B, only one concentration tested, I'd like to see a dose response.

We have completed a dose response and have included these new data as a replacement of Figure 1B. Thank you for this suggestion.

5. Fig. 1C did not rigorously prove that Cmpd A inhibits PD-L1/PD-1 signaling, because they did not exclude the possibility that Cmpd A works through other signaling pathways or some pleiotropic mechanisms. For example, hydrophobic compounds may induce cell clustering, which is not evaluated in the current manuscript. I also feel that a PD-1 KO or PD-L1 KO condition is required. If the authors' model is correct, then they should detect no effect of Cmpd A under either condition. Alternatively, they could test the effect of Cmpd A in the presence of saturating concentrations of anti-PD-L1 or anti-PD-1.

All good points and suggestions. To test for non-specific stimulation as the reviewer suggested we did two different experiments. In the first experiment, we used the CHO-PD-L1 KO cell line to test for non-specific activation of Jurkat PD-1 luciferase. While we do see some slight increase when Jurkat PD-1 cells are treated with cmpd A or cmpd B, this minor increase is not responsible for the significant luciferase activity we see specifically with cmpd A in the PD-1/PD-L1 bioassay as we do not see activation with the inactive cmpd B (Sup Fig 4A). We also went one step further and tested for general "non-specific" effects on immune cell signaling in fresh human PBMCs. Human PBMCs isolated from a healthy volunteer were treated with α PD-1, α PD-L1 antibodies, compound A, compound B, or LPS followed by measurements of cytokine release (22 cytokines tested). No non-specific effect on human PBMC cytokine release was observed for antibodies or compound A (Sup Fig 3C). We believe these data suggest the activity observed in Fig 1C is a result of a loss of PD-1/PD-L1 interaction and not a non-specific effect of compound A.

(page 7, line 130-132) Added text: In addition, non-specific activation of the Jurkat PD-1 luciferase or primary immune cells by compound A was not observed (Supplemental Figure 3C, 4A).

6. It was recently shown that PD-L1 interact with CD80 in cis to affects both PD-1 and CTLA-4 pathways. If Cmpd A induces PD-L1 homodimerization, how does it affect PD-L1/CD80 interaction, and how would this impact their interpretation of their results?

Indeed, recent evidence has shown CD80 to interact in *cis* with PD-L1 to affect both PD-1 and CTLA-4 signaling. While this manuscript was focused on the role of small molecule inhibitors in modulating the dynamics of the PD-1/PD-L1 axis, we suspect that cmpd A-induced dimerization and internalization would also impact CD80 cell surface interactions with PD-L1 for two reasons. First, we have mapped the CD80/PD-L1 interacting residues and find these residues would not be available with a compound A-induced PD-L1 dimer. Secondly, the subsequent rapid internalization of PD-L1 by compound A (Fig 4C) would certainly prevent cell surface CD80 interactions. Future studies are on ongoing to further delineate the role of cmpd A-induced dimerization and internalization on CD80 as well as other signaling molecules and will be the topic of a future upcoming manuscript.

7. The manuscript is concisely written, but the authors can better describe the methods and materials. As it stands, there is not enough information on how each experiment was conducted. For example line 411, why using anti-His6?

We have added more detail to the methods section. On the topic of anti-His6, mAb anti-6HIS Tb cryptate Gold (Cisbio) recognizes and binds to

recombinant PD-L1 protein containing His tag, which delivers energy to Pab antibody attached to recombinant PD-1 Fc protein. In order to be clearer, we have added the following to the methods section.

(page 23, line 442-444) Recombinant Human PD-L1-His (R&D systems) at 6 nM, small molecules and antibodies were added to 384 well plate (Corning) followed by recombinant Human PD-1 Fc Chimera (R&D systems) at 6 nM.

Reviewer #2 (Remarks to the Author):

In the manuscript, Park and co-workers present the discovery and biological activity of a symmetric small molecule inhibitor of PD-1/PD-L1 interaction. The authors also propose a unique mechanism of action of this inhibitor, which acts by inducing dimerization and internalization of PD-L1 in model CHO cells.

The manuscript is well written and the results are presented in an easy-to-follow form. The findings provide interesting mechanism of action of the molecule and contribute well to the progress in the field of targeting PD-L1 with small molecules. The reviewer suggests considering the publication of the manuscript, provided that the authors address the following (minor) questions/comments, which include lack of several critical controls in some of the experiments. Addressing these concerns would greatly increase the relevance of the conclusions drawn by the authors.

We would like to thank Reviewer #2 for their thoughtful review and kind remarks on our manuscript. We have addressed all reviewer #2 comments and feel this has greatly strengthened this manuscript.

1. Figure 1C – is it truly a % inhibition? How many repeats were done?

We agree with the reviewer that these units can seem a bit confusing. In this experiment, "% inhibition" indicates the blockade of PD-L1 interaction with PD-1, which results in increased NFAT-RE luciferase activity in Jurkat-PD-1 cells. Without PD-1/PD-L1 blockade using a small molecule or antibody, NFAT-RE luciferase reporter activity would remain suppressed in the Jurkat PD-1 target cells. 100% inhibition is full inhibition of this bioassay by 5 µg/ml αPD-L1 antibody (clone MIH1). Percent inhibition by cmpd A and cmpd B is plotted as "%inhibition" relative to this maximal inhibitory αPD-L1 antibody response following subtraction from the media control. The specific formula is as follows: $((\text{average cmpd RLU} - \text{media RLU}) / (\text{RLU at 5 } \mu\text{g/ml } \alpha\text{PD-L1 antibody} - \text{media RLU}))$. This experiment was repeated 3 times.

2. Figure 2A,B – how is the PD-L1 detected in whole cell lysates prepared with the M-PER reagent?

This is a good question. The manufacturers instructions for M-Per WCL extractions is to incubate for 15mins and pipet up and down. We tried this and indeed it was difficult under native conditions to get clear bands in this manner. So, we experimented with the best conditions and found that the NativePage (M-Per + 10%DDM) on the shaker for 1hr at 4°C allowed for better resolution of both the monomer and dimeric native forms of PD-L1. We have added some language to the methods to be clearer about this procedure.

Text added to methods section: (page 24, 461-487) PD-L1 aAPC/CHO-K1.....

3. Figure 2C, Figure 4A – flow cytometry: please clearly indicate the clones of antibodies that were used in the experiment. From the previous and following sections it can be assumed that MIH1 was used as an anti-PD-L1 control, and 29E.2A3 might have been used for FC. If this was the case, please analyze the possibility that the binding surface of 29E.2A3 overlaps with the binding surface of compound A and the second PD-L1 monomer. This would limit the binding of 29E.2A3 to PD-L1 in the presence of compound A, resulting in a lower FC signal.

We have included some language in the methods section for Fig 2C, 4A to be clearer about our protocol and PD-L1 detection methods for these two experiments. In Figure 2C & 4A, we used an acid wash protocol to clear the molecules being tested for internalization (MIH1 or compounds) from cell surface PD-L1. Detection of cell surface PD-L1 was performed with the same MIH1 clone used for testing internalization, but only after clearing this with the acid wash. As a control experiment to prove the acid wash could clear antibody or compound, we tested without the acid wash. Without acid wash, MIH1 PE Ab did not stain surface PD-L1 due to the pre-occupancy of PD-L1 by purified MIH1 Ab. We did not use clone 29E.2A3 in this study.

4. Figure 2D – what does the phrase “Images are representative of three experiments” mean? How many individual cells were visualized in each experiment? Figures 2 F and G seemingly show quantified data from a similar experiment. How many cells were monitored in this experiment? Error bars are extremely low (G) or absent (F) – was the data reproducibility really so high? Please explain.

“Images are representative of three experiments” means the data presented are similar in all three independent experiments. In addition, we looked at

>50 cells for each experiment in Figure 2D and have now notated this in the legend. Apologies for the confusion around Figures 2F/2G, in these experiments cell surface PD-L1 was measured by flow cytometry and not confocal. We have made this clear in the legends section for this Figure 2F/2G. For the error bars in Figure 2G these error bars are small because they represent technical replicates within a single experiment, but we have repeated this experiment and see the same results.

(Legend page 1, line 20-21) Changed text to: Images are representative of three experiments and a minimum of 50 cells observed in each experiment.

(Legend page 2, line 31-32) Changed text to: Red line represents reconstitution of cell surface PD-L1 as measured by flow cytometry of $\geq 10,000$ cells per time point.

5. Figure 2E: the Co-IP experiment lacks necessary controls: the detection of Myc in precipitates is missing (the control of equal cMyc-PD-L1 amounts in DMSO and CmpdA eluates) and detection of Flag in input samples (the control of equal PD-L1-Flag amounts in DMSO- and CmpdA-treated cell lysates).

Apologies for this oversight, we have repeated this experiment with all proper input controls and included these data as a replacement for Figure 2E.

6. Figure 4B: why were the compound B-treated cells considered a control for the experiment and not the untreated cells?

Compound B was used as a control in this experiment to control for any potential non-specific compound related stimulation of T cell function. We did not see any effect of compound B on T cell function when compared to media only control, so we thought it best to include this as a “chemotype” specific control that has no activity against PD-L1.

7. Lines 263-264: “We believe this is the first report of a cellular potent low molecular weight small molecule PD-1/PD-L1 checkpoint inhibitor” – please refer to the previous manuscripts (Basu et. al 2019, DOI: 10.1021/acs.jmedchem.9b00795, and Skalniak et. al 2017, DOI: 10.18632/oncotarget.20050), where bioactive small molecules targeting PD-L1 were characterized.

Indeed, both articles are very important and have been cited in this manuscript. While we do feel our report is the first “comprehensive” characterization of a low molecular weight PD-L1 small molecule inhibitor in cells, animal models and human samples, we agree that perhaps the statement made in the current version of this manuscript is a bit strong and have therefore excluded this sentence.

Reviewer #3 (Remarks to the Author):

In the present study Park et al. describe the properties of a novel compound (named compound A) that induced internalization of PD-L1 resulting in suppression of PD-1:PD-L1 interaction, and its implications in T cell activation and anti-tumor function. The authors propose that use of this compound would have equivalent effects with antibodies blocking that PD-1 pathway and could potentially substitute for the use of such antibodies for clinical applications. Although the data are of potential interest several points require further investigation before conclusions can be made.

We would like to thank Reviewer #3 for their helpful and thoughtful review and have addressed all reviewer #3 suggestions. We feel this has greatly strengthened this manuscript.

Major points:

1) Important details on the HTRF assay developed by the authors are missing. This should be described in a comprehensive manner and the interpretation of the relevant results using this approach should be thoroughly outlined.

We thank the reviewer for this suggestion and have added more details on the HTRF including more interpretation of the relevance of these results.

2) Compound A blocks PD-L1 interaction with PD-1. Does it also block PD-L1 interaction with B7-1?

Indeed, recent evidence has shown CD80 to interact in *cis* with PD-L1 to affect both PD-1 and CTLA-4 signaling. While this manuscript was focused on the role of small molecule inhibitors in modulating the dynamics of the PD-1/PD-L1 axis, we suspect that cmpd A-induced dimerization and internalization would also impact CD80 cell surface interactions with PD-L1 for two reasons. First, we have mapped the CD80/PD-L1 interacting residues and find these residues would not be available with a compound A-induced PD-L1 dimer. Secondly, the subsequent rapid internalization of PD-L1 by compound A (Fig 4C) would certainly prevent cell surface CD80 interactions. Future studies are on ongoing to further delineate the role of cmpd A-induced dimerization and internalization on CD80 as well as other signaling molecules and will be the topic of a future upcoming manuscript.

3) The authors stated that ligand induced dimerization has been reported for surface receptors and use this as a justification to study the effects of compound A on PD-L1 dimerization. Although the nature of compound A is not disclosed, do they imply that compound A is a natural PD-L1 ligand that might mediate ligand induced dimerization? Without providing information

whether compound A is a natural partner of PD-L1 this justification is scientifically inaccurate.

This is a good point raised by the reviewer. Compound A structure has been provided in Supplemental Figure 2A. Compound A is a symmetrical molecule that interacts with two PD-L1 proteins through a unique hydrophobic pocket created between antiparallel β -sheets comprised of symmetrical tyrosine residues which π -stack with compound A (Sup Fig 2B). The core bi-phenyl in compound A is critical for the positioning of compound A to allow for π -stacking with PD-L1 dimer. Compound A was synthesized in the laboratory specifically to interrogate the potential to dimerize PD-L1 proteins through these unique interactions, therefore compound A is not a natural ligand found within an organism. While we cannot rule out that there may exist some yet undiscovered natural ligand capable of inducing this PD-L1 dimerization and internalization, the current manuscript is focused on pharmacologically-induced PD-L1 dimerization as a potential starting point for a novel small molecule PD-L1 inhibitor with this unique mechanism of action.

4) The authors support that compound A induced PD-L1 dimerization and subsequent internalization. There are no appropriate experimental data to support this conclusion. The assays shown using native gel electrophoresis do not provide evidence of the mechanism involved. Specific methods are available to assess molecular dimerization at the cell membrane and such assays should be employed.

We respectfully disagree with the reviewer that our data do not support compound A-induced PD-L1 dimerization. We have seen compound A-induced dimerization in solution within the homodimer HTRF assay (Fig 1B), and we have seen compound A-induced dimerization in our crystallographic studies. We further extended this finding to the cell by treating cells with compound A and detecting PD-L1 in its native form (Fig 2A). We see a doubling of PD-L1 size when cells are treated with compound A (Sup Fig 3B) and when we treat just lysates with compound A we do not see PD-L1 dimerization suggesting this dimerization is occurring in intact cells on the cell membrane (Fig 2B). Furthermore, when we test for the temporal aspect of compound A-induced PD-L1 internalization, we see that this compound A effect on cell surface PD-L1 occurs very rapidly within 5 mins (Sup Fig 4C). In addition, we can co-immunoprecipitate myc-tagged PD-L1 with flag-tagged PD-L1 only when cells are treated with compound A (Fig 2E). We feel strongly that the preponderance of our data suggest compound A induces dimerization of PD-L1 in living cells and this likely triggers the internalization step. Further studies will be focused on the exact biochemical mechanism of this compound A-induced internalization.

5) Figure 2A: The investigators used cell lysates to assess the effects of compound A on PD-L1 by native PAGE The usual approach by which proteins

are assessed after native PAGE is Coomassie staining. Is this what they did? There is no information how the authors assessed the proteins after electrophoresis. Cell lysates contain multiple proteins and the identity of the bands shown in the gels is unclear. Two separate bands are present at the area at which the authors indicate "PD-L1 monomer" but their identity is uncertain. CHO cells that do not express PD-L1 should also be used as control in this assay.

We have included more details of the detection methods deployed in the native gel electrophoresis studies in the experimental methods section as outlined below. In addition, we have included a native gel with protein markers (Sup Fig 3B) and as expected the lower band is the known size of monomeric PD-L1 (~75kDa) which doubles in size upon compound A treatment (~150kDa). The doublet seen with monomeric PD-L1 is consistent with the known glycosylation forms of monomeric PD-L1 (Chia-Wei, Li et al Nature Communications, 2016). We have included this reference and mention of this doublet in the paper. In addition, we agree with the reviewer that CHO-PD-L1 KO cells would be the best control and have completed these experiments. We see no PD-L1 staining in the CHO PD-L1 KO suggesting the bands we see are indeed specific for PD-L1. This blot has been added to the manuscript (Supplemental Fig 5B).

(Page 24, line 461-487) PD-L1 aAPC/CHO-K1.....

6) It is unclear whether compound A induces specifically PD-L1 dimerization or other cell surface proteins are involved in the observed effects. To clarify this, it is necessary to use purified PD-L1 protein to assess whether compound A can induce dimerization.

We thank the reviewer for this suggestion and have completed this experiment and included these data in the manuscript (Sup Fig 4B). Indeed, we can induce dimerization of recombinant PD-L1 with compound A treatment as measured by western blot. We have also added language in the text to address this new experimental finding as follows

(Page 18; 334-337): "In addition, we see dimerization of recombinant PD-L1 when treated with compound A suggesting the dimerizing effect of compound A on PD-L1 is direct and does not require other host proteins."

7) Figure 2C: The authors claim that treatment with compound A for 1 hour results in dimerization and internalization of PD-L1 leading to loss of surface PD-L1 expression as assessed by flow cytometry. It is necessary to perform detailed kinetics of PD-L1 expression levels on cell surface and cytoplasm by flow cytometry during multiple time points of treatment with compound A, to accurately assess changes on PD-L1 expression and subcellular localization.

Great idea! We have completed this experiment and included in Supplemental Figure 4C. As expected, we see that with as little as 100nM compd A induces internalization of cell surface PD-L1 within 5 mins.

8) Figure 2C: Which antibody clones were used for anti-PD-L1 staining? No specific information is provided. The PD-L1 Ab Biolegend #329724 is clone 29E.2A3 PD-L1 Ab, which blocks both PD-1 and B7-1 interactions with PD-L1. Since compound A also blocks the PD-1-PD-L1 interaction, this antibody should not be used to detect PD-L1 because potentially compound A interferes with staining. It is necessary to use multiple anti-PD-L1 antibody clones to assess parallel surface and cytoplasmic expression of PD-L1 after treatment with compound A or control.

We have included some language in the methods section for Fig 2C, 4A to be clearer about our protocol and PD-L1 detection methods. For example, in Figure 2C & 4A, we used an acid wash protocol to clear the molecules being tested for internalization (MIH1 or compounds) from cell surface PD-L1. Detection of cell surface PD-L1 was performed with the same MIH1 clone used for testing internalization, but only after clearing this with the acid wash. As a control experiment to prove the acid wash could clear antibody or compound, we tested without the acid wash. Without acid wash, MIH1 PE Ab did not stain surface PD-L1 due to the pre-occupancy of PD-L1 by purified MIH1 Ab. We did not use clone 29E.2A3 in this study.

Added text (page 31 605-616) Acid-wash internalization assay PD-L1 aAPC/CHO-K1 cells were incubated with small molecules or antibodies (purified aPD-L1 antibody clone MIH1, or Opdivo aPD-1 antibody) for 1 hour and washed with 1x PBS three times. Some cells were further washed with acid wash buffer (DMEM, 0.2 % BSA, pH 3.5) three times for five minutes on the shaker to strip off the small molecules and antibody bound to PD-L1 on the cell surface, then another three washes with 1x PBS. Lift buffer (10 mM Tris, 140 mM) was used to detach the cells from the cell plate. Cells treated with or without acid wash were stained with anti-PD-L1 antibody (clone MIH1, Invitrogen) and Live/Dead™ fixable Aqua dead cell stain kit (L34965, ThermoFisher). Cells were acquired by LSRFortessa (BD Bioscience, CA) and analyzed by Flowjo (TreeStar, OR).

9) The authors claim that treatment with compound A results in dimerization and internalization of PD-L1 but no such effect was observed after treatment with anti-PD-L1 or anti-PD-1 antibodies. Antibodies are bivalent and can dimerize PD-L1 as well, so why anti-PD-L1 antibody incubation does not cause internalization and loss of PD-L1 surface expression?

This is a good question. While α PD-L1 antibodies are indeed bivalent, we do not see induction of dimerization with antibodies as measured by homodimer HTRF or native gel (Fig 1B; Sup Fig 5B). Future experiments are currently focused on mutational studies designed to identify the specific triggers and mechanisms of internalization by compd A and will be the topic of a future publication.

10) Figure 3A: Tumor experiment. Compound A was administered for 7 days and outcomes on targeted populations were assessed several days later (on day 28). According to the results shown in Figure 2F, wash out of compound A resulted in re-expression of PD-L1 to baseline levels. Based on these data, it is obvious that after cessation of in vivo administration, decrease of compound A levels will result in gradually diminished efficacy and recovery of PD-L1 expression on the targeted cells. Figure 3B shows decreased levels of PD-L1 expression on tumor CD45+ cells on day 28. It is not feasible to interpret these results without information about the pharmacokinetics and the clearance of compound A after in vivo administration. Furthermore, PD-L1 expression on target populations should be assessed at multiple time points after in vivo treatment.

We thank the reviewer for this suggestion and the revised Supplemental Figure 5A now includes assessment of compound A concentrations present in the tumors at day 28. As can be seen in the new data, compound A is still present at levels ranging from 0.6 to 4.1-fold the EC50 in tumors, even at 21 days post-last dose administration, which may account for the durable efficacy observed. Furthermore, we have added a correlation graph (Fig 3C) indicating that animals with the greatest levels of compound A present at day 28 also had the greatest tumor inhibition response, providing additional support for a role of compound A dose levels in correlating with the efficacy profile observed at study end.

Regarding the comment that PD-L1 expression should be assessed at multiple timepoints after treatment, we thank the reviewer for this suggestion. This is one element that will be assessed in planned follow-up studies; the study described in the current manuscript represents an initial *in vivo* proof-of-concept and subsequent work will examine the time course of PD-L1 reduction as well as optimal compound A dose and treatment schedule.

11) The effects of compound A on PD-L1 expression on tumor cells should also be examined.

We apologize for the error, but the original data has been confirmed as pertaining to CD45⁻ tumor cells (not CD45⁺ cells as originally stated). This has now been corrected in the text and figure.

12) After analyzing the numbers of CD3⁺, CD4⁺ and CD8⁺ cells circulating in the blood, the authors concluded that the effects of compound A on these cell populations were similar to those induced by anti-PD-L1 antibody. However, the data show that compound A had only a slight effect on CD4⁺ T cells and no effect on CD8⁺ or total CD3⁺ T cells. In contrast, anti-PD-L1 antibody treatment resulted in significant expansion of all these T cell populations. Thus, these conclusions are inconsistent with the experimental data.

We acknowledge that the effects of compound A on CD4⁺ and CD8⁺ T cells are not as significant as that observed with the α PD-L1 antibody treatment. However, the trend observed with compound A treatment is similar, though not statistically significant for the effect on CD8⁺ or CD3⁺ T cells. Further studies to optimize compound A dose level and treatment schedule may result in more significant effects on T cell expansion, though for the initial *in vivo* proof-of-concept described in this manuscript we believe the trends are suggestive of a similar effect as α PD-L1 antibody treatment.

13) Line 123: Anti-PD-1 antibody is mentioned here instead of anti-PDL1 that is shown in the figure.

Text changed to: (page 7, line 128) Interestingly, both compound A and α PD-L1 antibodies exhibit similar upregulation of...

REVIEWERS' COMMENTS

Reviewer #1 (Remarks to the Author):

In general, the authors have done a good job addressing my comments, except that whether CompA inhibits B7-1/PD-L1 interaction. I feel that if they choose not to show the data due to another publication, this issue still deserves to be mentioned in the discussion section.

Reviewer #2 (Remarks to the Author):

The authors addressed all of my concerns. The manuscript is ready for publishing.

Reviewer #3 (Remarks to the Author):

The authors have addressed my previous concerns, performed additional work and revised their manuscript accordingly. I do not have any further questions and I think the manuscript is acceptable for publication in the present form.